



# Observing the timescales of aerosol-cloud interactions in snapshot satellite images

Edward Gryspeerdt[1], Tom Goren[2], and Tristan W. P. Smith[3]

[1]Space and Atmospheric Physics Group, Imperial College London, UK
[2]Institute for Meteorology, Universität Leipzig, Germany
[3]UCL Energy Institute, University College London, UK

**Correspondence:** Edward Gryspeerdt (e.gryspeerdt@imperial.ac.uk)

**Abstract.** The response of cloud processes to an aerosol perturbation is one of the largest uncertainties in the anthropogenic forcing of the climate. It occurs at a variety of timescales, from the near-instantaneous Twomey effect, to the longer timescales required for cloud adjustments. Understanding the temporal evolution of cloud properties following an aerosol perturbation is necessary to interpret the results of so-called "natural experiments" from a known aerosol source, such as a ship or industrial

site. This work uses reanalysis windfields and ship emission information matched to observations of shiptracks to measure the timescales of cloud responses to aerosol in instantaneous (or"snapshot") images taken by polar-orbiting satellites.

As found in previous studies, the local meteorological environment is shown to have a strong impact on the occurrence and properties of shiptracks, but there is a strong time dependence in their properties. The largest droplet number concentration ($N_d$) responses are found within three hours of emission, while cloud adjustments continue to evolve over periods of ten hours

or more. Cloud fraction is increased within the early life of shiptracks, with the formation of shiptracks in otherwise clear skies indicating that around 5-10% of clear-sky cases in this region may be aerosol-limited.

The liquid water path (LWP) enhancement and the $N_d$-LWP sensitivity are also time dependent and strong functions of the background cloud and meteorological state. The near-instant response of the LWP within shiptracks may be evidence of a retrieval bias in previous estimates of the LWP response to aerosol derived from natural experiments. These results highlight

the importance of temporal development and the background cloud field for quantifying the aerosol impact on clouds, even in situations where the aerosol perturbation is clear.

## 1 Introduction

The response of a cloud to an aerosol perturbation is fundamentally time-sensitive. Increasing the number of cloud condensation nuclei (CCN) increases the number of cloud droplets at cloud base (Twomey, 1974) almost immediately, resulting in

an instantaneous change to the properties of an individual air parcel. The $N_d$ and effective radius ($r_e$) in the rest of the cloud respond on a timescale related to the cloud geometrical depth and the in-cloud updraught (on the order of 10-20 minutes for a cloud thickness of 200m and an updraught of $0.2\,\mathrm{ms}^{-1}$). This response is referred to as the Twomey effect, which leads to the radiative forcing from aerosol-cloud interactions (RFaci; Boucher et al., 2013).




Changes in droplet size can also impact precipitation processes, leading to further changes in liquid cloud properties notably the liquid water path (LWP) and cloud fraction (Albrecht, 1989, e.g.). Further changes to the LWP and CF may come through aerosol-dependent entrainment and mixing processes (Ackerman et al., 2004; Xue and Feingold, 2006; Bretherton et al., 2007; Seifert et al., 2015). The timescale for these processes is longer than the $N_d$ timescale, as they proceed through a modification

of process rates, requiring several hours to generate a significant change in the LWP (Glassmeier et al., 2020). For a large-scale change in cloud amount, the timescales may be even longer, as it requires the switching of cloud regime, from open to closed celled convection (Rosenfeld et al., 2006; Goren and Rosenfeld, 2012). The timescale for this aerosol response is related to the timescale for a switch between open and closed cells and related to the lifetime of individual cells (around 2 hours Wang and Feingold, 2009a).

One of the largest uncertainties when using observations to constrain aerosol-cloud interactions is the impact of meteorological covariations, where aerosol and cloud properties are both correlated to the same meteorological factors (such as relative humidity). Variations in this factor will then generate relationships between aerosol and cloud properties, even without a causal impact of aerosol on cloud (e.g. Quaas et al., 2010). Although a number of methods for identifying causal relationships have been proposed (e.g. Koren et al., 2010; Gryspeerdt et al., 2016; McCoy et al., 2020), "natural experiments" (Rosenzweig and

Wolpin, 2000), where the aerosol is perturbed independently of meteorology (such as by ships or industry Conover, 1966; Toll et al., 2019) are the standard way of isolating the causal aerosol effect. As these studies are often performed at relatively short times after the aerosol perturbation, understanding the timescales of the response is essential to interpret these results.

The majority of satellite observations provide a static picture of the Earth, limiting their ability to characterise liquid cloud temporal development. Previous studies have addressed this by using multiple observations to build a composite diurnal cycle.

Matsui et al. (2006) showed that the MODIS diurnal cycle is correlated to the aerosol environment, with a clear importance of the initial cloud state. Meskhidze et al. (2009) and Gryspeerdt et al. (2014) showed that short term development is also correlated to the aerosol environment, but that accounting for the initial cloud state is vital. This technique has recently been extended to longer timescales (Christensen et al., 2020).

These studies have focus on existing variability in aerosol. This allows cloud temporal development to be investigated at a

global scale, but limits the ability to measure timescales directly. Exogeneous aerosol perturbations (such as those from ships) are emitted independently of meteorological factors. With a limited spatial extent, they have clearly identifiable polluted and control regions, allowing the impact of the aerosol on the cloud field to be inferred (Durkee et al., 2000a), vary along their length (Kabatas et al., 2013) and can be tracked over several days, providing evidence of a long-lasting aerosol perturbation to cloud properties (Goren and Rosenfeld, 2012).

In this work, the time development of shiptracks is used to quantify the timescales of aerosol-cloud interactions in marine boundary layer clouds and how these are affected by meteorology. Although shiptracks appear in satellite images as linear cloud formations, they have no ability to transmit information along their length. This means that they can be considered as a chain of independently perturbed clouds with a similar initial aerosol perturbation (Kabatas et al., 2013). By linking shiptracks identified in satellite images (e.g. Segrin et al., 2007) with ship $SO_x$ emissions estimated from transponder data (Gryspeerdt et al., 2019b),

the aerosol perturbation is identified independently of the cloud properties. The ship motion and reanalysis windfield are used





to estimate the emitted aerosol trajectory, enabling the conditions controlling shiptrack formation to be identified and providing a time-axis for the perturbation. This enables the measurement of aerosol impacts on cloud over a twenty hour time period for each ship (approximately equal to the longest identified shiptrack). This work uses this snapshot method for measuring time dependence to investigate the role of meteorological properties in development of the macrophysical (length, detectability and

width) and microphysical properties ($N_d$, LWP) shiptracks, their sensitivity to the aerosol perturbation, and what this means for the potential radiative forcing.

## 2   Methods

### 2.1   Shiptrack locations

This work uses the shiptrack and ship locations from Gryspeerdt et al. (2019b). The shiptracks were logged by hand in MODIS

Aqua day microphysics images (Lensky and Rosenfeld, 2008), with the $N_d$ (following Quaas et al., 2006) used in ambiguous cases. These shiptracks are linked to individual ships using ship AIS transponder locations, with emissions estimated using this AIS information and the ship physical properties (Smith et al., 2015). All of the tracks used are linked to ships from 2015 in a region off the coast of California (30-45N, 115-130W).

For each ship, the estimated trajectory of the emitted $SO_x$ is determined by advecting the historical ship positions with the

1000 hPa reanalysis windfield from ERA5 (as in Gryspeerdt et al., 2019b). This level was selected as it provided the best match between the trajectories and the observed shiptrack locations, likely due to the thin boundary layers in this region. For the observed shiptrack, the time since emission is then determined as the time since emission at the closest point on the emission trajectory. This emission trajectory is also used to extend the observed shiptrack to 20 hours since emission.

Ship location data has to be interpolated between sparse AIS observations, which can lead to significant error in the locations.

A comparison with ship meteorological reports suggests that this interpolation error can often be as large as 100 km in the ship position, compounding further in the estimated emission trajectories. To avoid this interpolation uncertainty, only cases where normalised Frechet distance between the reconstructed plume and the identified shiptrack less than 0.5 are included to ensure an unambiguous match between the ship and shiptrack (following Gryspeerdt et al., 2019b). This leaves 1,209 shiptracks for use in this study.

### 2.2   Identifying polluted regions

The shiptrack identification method is based the method from Segrin et al. (2007) and Christensen et al. (2009), with modifications outlined in Gryspeerdt et al. (2019b). The identified track is divided into 10 pixel long chunks (each MODIS pixel is 1 km across at nadir, rising to over three at the swath edge). Within each chunk, pixels are classed as "detected" when they are more than 2 standard deviations above the background (the standard deviation excludes detected pixels). The detected pixels are

grouped with nearby pixels; groups far from the track central location are classed as "polluted (non-track)" and excluded from this analysis. Fig. 1a shows an example, with the detected shiptrack in the centre and a second shiptrack at the edge that has

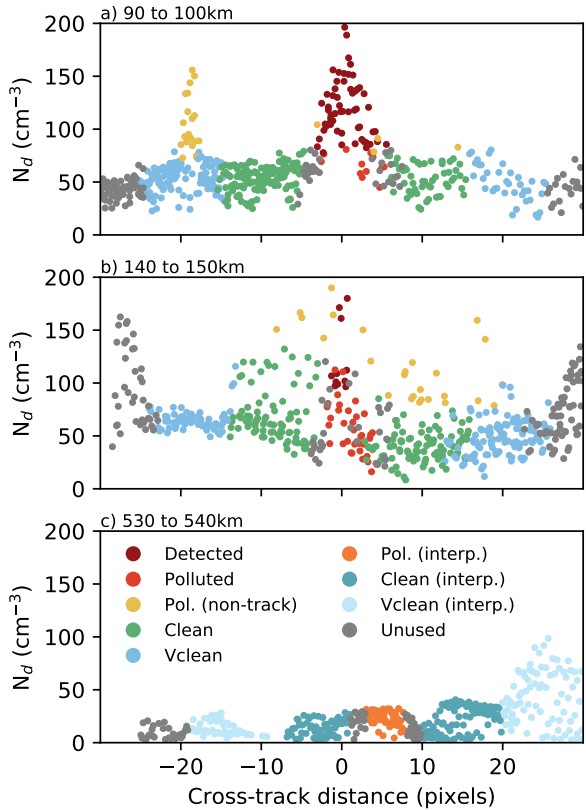

**Figure 1.** Example track chunks and the identification of polluted pixels along the track shown in Fig. 2. Marker colours are shown in (c). Distances in the titles are from the ship location at the time of the satellite overpass. "Detected" pixels are those located by the Segrin et al. (2007) algorithm. "Polluted" pixels are added to the in-shiptrack region by the modifications in this work. "Polluted (non-track)" pixels were detected, but removed from the shiptrack region in this work. The "clean" pixels are used as an unperturbed control situation, while "vclean" is a further control. The interpolated pixels (c) are from segments where no detected pixels exist. Further details are in the text.

been excluded due to being too far from the hand-identified shiptrack location. The edges of the shiptrack within each chunk are defined by the furthest spaced detected pixels, with extra pixels in that region being classed as "polluted". This polluted regions is used to compile the statistics in this work.

With a buffer region of two pixels ($>2$ km), corresponding clean/control regions are identified 10 pixels either side of the track and "vclean" regions a further 10 pixels either side of that. Distances are in pixels to keep an approximately similar number of pixels in each region for each segment. The clean and vclean regions exclude the polluted (non-track) pixels, as nearby shiptracks (as shown in Fig. 1a) are not representative of the background cloud state. This may not always be the correct decision, as crossing shiptracks (Fig. 1b) may be more representative of the background. This decision has a negligible impact on the results presented in this work.



Not all segments contain detected pixels. In this case, the width and cross-track location of the shiptrack are interpolated from the nearest segments with detected pixels (Figs. 2, 1c).

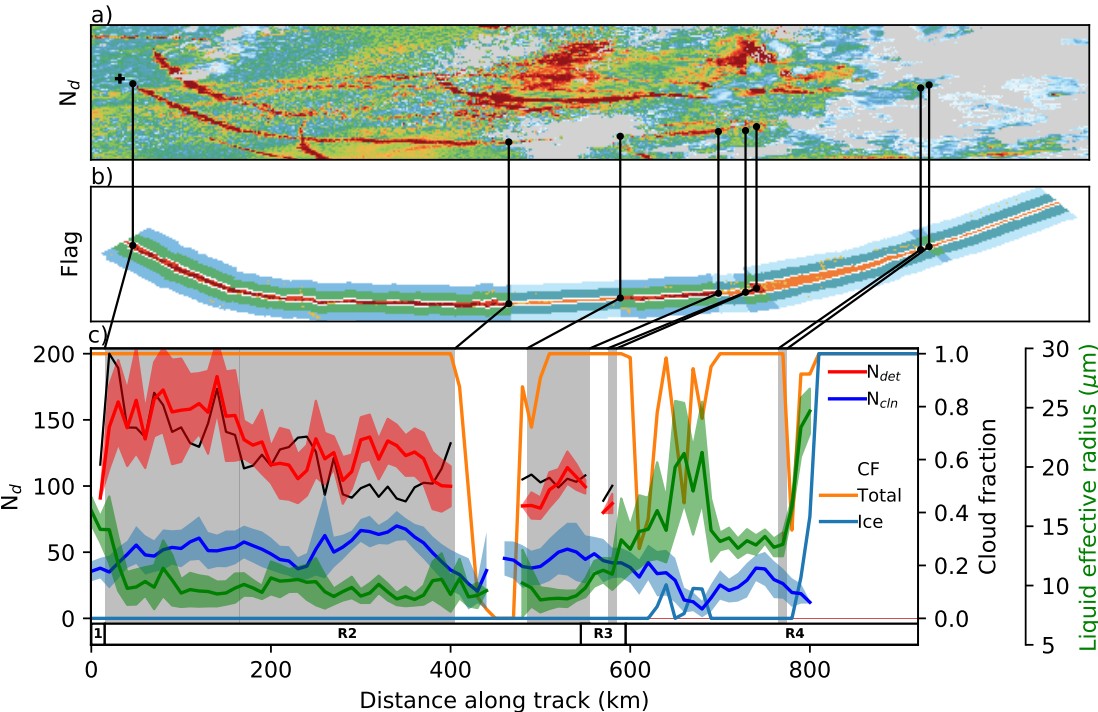

**Figure 2.** An example shiptrack. a) The $N_d$ field from MODIS, grey regions indicate no liquid cloud. The ship position is marked with across and the dot markers are to identify regions of the shiptrack. b) Pixels as identified by the shiptrack algorithm. Red regions are polluted regions of the shiptrack and green the corresponding clean regions. Orange regions are interpolated shiptrack locations, based on the identified polluted regions as described in the text. Blue regions are the corresponding unperturbed regions for the interpolated shiptrack locations. c) The change in properties along the shiptrack. Grey regions are "detected" shiptrack segments (polluted pixels are identified) and the black lines connect the edges of these regions between subplots. The black bar along the top shows the section of the track identified by a human from the day-microphysics imagery. Note that these do not have to overlap. Red is $N_{pol}$ for regions where the shiptrack is detected and blue $N_{cln}$ with the shaded regions showing the standard deviation within that segment. The thin black line is the $N_d$ enhancement, $\varepsilon_N$ ($N_{pol}/N_{cln}$). The unperturbed effective radius and standard deviation are shown in green. The total cloud fraction is in orange and the ice cloud fraction in light blue.

The classifications are then aggregated into 15 minute (10 km) segments. This is a short enough time period to allow the initial development of the track to be resolved and these two measures are approximately equal for relative windspeed of
5    40 kmh$^{-1}$ (Durkee et al., 2000a). A full 20 hour shiptrack classification is shown in Fig. 2b, with the flag colours following Fig. 1. Note that polluted pixels are identified even in segments where there are no detected pixels (a white background in





Fig. 2c), even within the hand-identified region (region R2 along the bottom of Fig. 2c) by using the extrapolated shiptrack location based on reanalysis winds.

This method identifies polluted pixels along the length of the shiptrack together with a matching unperturbed region outside the shiptrack. With these regions identified for each shiptrack, the cloud properties along the length of each shiptrack are

registered (Fig. 2c). The $N_d$, for polluted (red; $N_{pol}$) and unperturbed (blue; $N_{cln}$) regions, is calculated following the method of Quaas et al. (2006), applying the filtering as outlined in Grosvenor et al. (2018). In this example, there is a general decrease in $N_d$ along the length of the shiptrack, but with a considerable fluctuation. The detected $N_d$ enhancement factor (thin black line; $\varepsilon_N = N_{pol}/N_{cln}$) shows a much smoother decrease, as it incorporates correlated fluctuations in $N_{pol}$ and $N_{cln}$.

The retrieved cloud properties in this work are from the MODIS Aqua collection 6.1 cloud product (MYD06L2; Platnick

et al., 2017) and the meteorological properties are from the ERA5 reanalysis. Supplemental information about the background aerosol comes from the sulphate concentration in the MERRA reanalysis, following McCoy et al. (2017).

The uncertainties throughout this work are calculated using a bootstrap method (Efron, 1979) with 1000 samples and are shown with a 16-84% uncertainty range. This range is chosen such that significance at an individual time between high and low SO$_x$ is indicated by the non-overlap of these ranges.

## 2.3   Potential radiative forcing

To compare the radiative effect of shiptracks in different conditions, a potential radiative forcing (PRF) is calculated following Eq.1. This is not the true radiative forcing of the shiptrack, as it ignores diurnal variations in the cloud response to aerosol and incoming solar flux (F$^{\downarrow}$; approximate by a constant 280 Wm$^{-2}$).

$$PRF = F^{\downarrow} A \left( (\alpha_{pol} - \alpha_{surf}) CF_{pol} - (\alpha_{cln} - \alpha_{surf}) CF_{cln} \right) \tag{1}$$

The PRF is calculated for each segment individually, with $A$ denoting the area of the polluted region within the segment. The cloud albedo ($\alpha$) is calculated from the cloud optical depth ($\tau$) following Eq. (2). The surface ocean albedo is taken as a constant 0.04 (a representative value from Jin et al., 2004). Only liquid clouds are considered in the CF. The impact of CF changes is calculated by setting $CF_{pol} = CF_{cln}$ to exclude the impact of CF adjustments.

$$\alpha = \frac{\tau}{\tau + 7} \tag{2}$$

## 3   Results

The results from this work are split into two sections, the first deals with the macrophysical properties of the shiptrack (length, width, detectability, cloud fraction), whereas the second focuses on the microphysical properties and the liquid water path.





## 3.1 Macrophysical properties

### 3.1.1 Track length

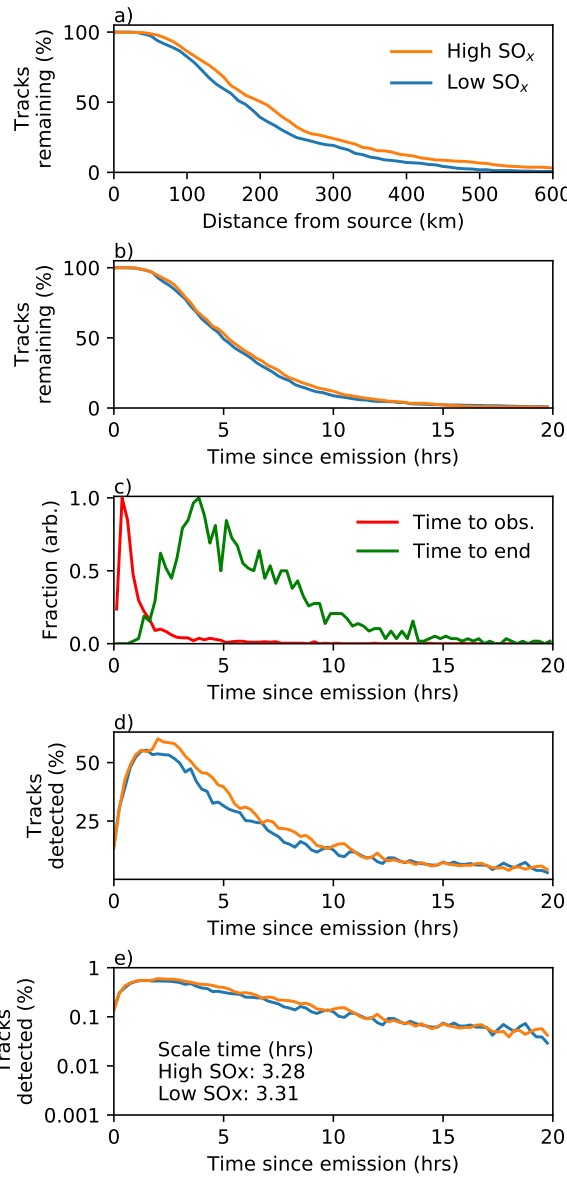

**Figure 3.** a) Track length (distance to oldest observed segment) for ships with high and low $SO_x$ emissions. b) The same as (a), but using the time since emission instead of the distance. c) histograms showing the time to first observation (red) and last observation (green) for the tracks used in this study. d) The percentage of tracks that are "detected" within any given 15 minute segment. Note that this is less than 100% in many cases due to gaps in shiptracks. e) as (d) but with a log y-axis.





The median shiptrack in this study is last observed at a distance of about 200 km from the source ship, with longer shiptracks being observed behind ships with higher $SO_x$ emissions (Fig. 3a). The difference is significantly reduced when using time as a coordinate (Fig. 3b). This is due to faster ships (producing longer shiptracks) typically burning more fuel and so emitting more $SO_x$. To account for this, we use time since emission as the along-track coordinate throughout the rest of this work.

These is a large variation in the time to initial observation (time to obs.) and the time to last observation for the shiptracks in this work (Fig. 3c). As in previous studies (Durkee et al., 2000b), the mean time to a first observation of the shiptrack is under an hour, with this database having a mean of 45 min. There is a long tail, with some tracks not being observed until five or ten hours since emission. Despite this long time period and the lack of a clear shiptrack "head", the advection of the ship emissions allows a clear identification and link to generating ship to be made in this work. A large variation in the length of the

observed section of the shiptrack is also found, with some shiptracks lasting only an hour and others lasting almost 20 hours, with a median length of 5 hours. Note that this distribution will vary between studies, as it depends on the criteria used to select the tracks.

Many of these shiptracks have gaps (segments where the track is not detected). The maximum detected fraction is around 1-2 hours after emission, where each 15 min segment has an approximately 60% chance of containing a detected shiptrack (Fig. 3d).

Although the difference is small, ships with higher $SO_x$ emissions produce shiptracks that are more likely to be detected for a longer period of time (Fig. 3d, e). This suggests that also the overall lifetime for the whole population of shiptracks is not a strong function of $SO_x$, ships with larger $SO_x$ emissions are more likely to generate tracks that can be detected for a longer period of time and so might be expected to have a larger radiative impact.

### 3.1.2   Track detection/formation

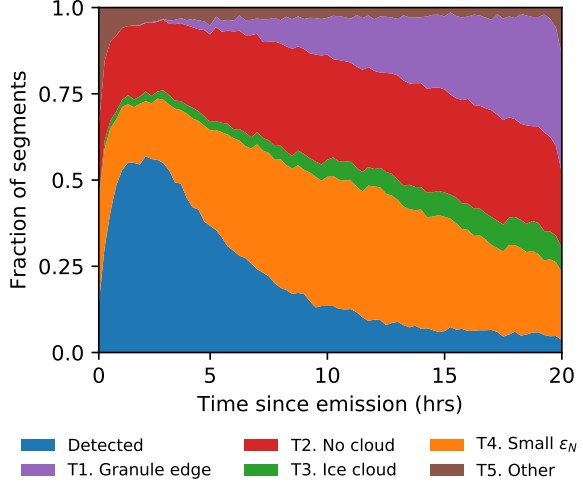

**Figure 4.** Reasons for track non-detection in segments as a function of time since emission. The tests are applied in the order T1-4.





The majority of segments do not contain a detected track; what limits shiptrack formation in these segments? For each segment lacking a detected track, four tests are applied in order: T1, is the segment outside a MODIS image (granule); T2, is there no cloud in the segment; T3, is the segment ice cloud and T4, is the $\varepsilon_N$ less than 1.4 (a 40% enhancement)? The percentage of segments satisfying these tests are shown in Fig. 4, with the tests being applied in order (so segments with satisfying T2 are
not checked for T3 or T4).

The The majority of shiptracks are located close to the edge of a MODIS image ("granule"), only a small number of shiptracks disappear because they reach the edge of the MODIS granule. Around 30% of shiptracks reach a granule edge after 20 hours after emission. In around 25% of segments, a lack of cloud prevents the detection of a shiptrack. This is expected given the cloud fraction in this region. Similarly, the small impact of overlying ice cloud on the detection of shiptracks is primarily due
to the low ice cloud fraction in subtropical subsidence regions.

The impact of cloud fraction on shiptrack detection is approximately constant with time since emission. In contrast, the impact of a small $\varepsilon_N$ increases with time since emission. Without the individual history of the segments, it is not possible to conclusively separate the dissipation of the shiptrack from an increase in the background $N_d$. The small $\varepsilon_N$ (T4) provides an indication of segments which could potentially form detectable shiptracks with increases in ship $SO_x$ emissions, although
meteorological conditions will also prevent the formation or observation of shiptracks in some conditions (Noone et al., 2000; Possner et al., 2018; Gryspeerdt et al., 2019b). Even in a region with large amounts of low-level liquid cloud, the frequency of occurrence of liquid cloud is a significant control on the length of shiptracks.

### 3.1.3   Track widths

Shiptrack width (defined as the maximum cross-track distance between polluted pixels within a segment) increases gradually
with time since emission (Fig. 5a), similar to previous studies (Durkee et al., 2000a). Although the shiptrack width is highly sensitive to outlier polluted pixels, a weak dependence of width on $SO_x$ emissions is observed, with the width of higher $SO_x$ emission shiptracks increasing faster with time.

This sensitivity of track width to $SO_x$ depends on the background cloud state. Considering only segments with a lower out-of-track CF ($CF_{cln}$<0.8), the width rises slowly over time, but with little sensitivity to the initial aerosol perturbation (Fig. 5b).
In contrast, the high $CF_{cln}$ segments have a width that is more sensitive to the ship $SO_x$ emissions (Fig. 5c).

Differences in the cloud regimes indicated by cloud fraction may explain this difference in behaviour. Lower $CF_{cln}$ situations are more likely to be open celled stratocumulus (Muhlbauer et al., 2014). In this situation, the width of the track is controlled primarily by the cell width, rather than the aerosol perturbation, due to a less efficient mixing of aerosol between cells (Scorer, 1987). Over time, mixing between the cells due to the collapse and creation of new open cells (Feingold et al., 2010) allows
the track to widen, gradually affecting a wider region as more cells are included in the track.

The high $CF_{cln}$ situation is more likely to be closed-celled stratocumulus or sea-fog (Ackerman et al., 1993). Particularly in the sea-fog case, the lack of a strong cellular structure control on the shiptrack width allows a dependence on the $SO_x$ emissions, with higher $SO_x$ emissions creating a wider plume and so a wider initial shiptrack. The quick and continued growth of the shiptrack in the first five hours for these high $CF_{cln}$ cases (Fig. 5c) therefore may be linked to this initial plume dispersion.





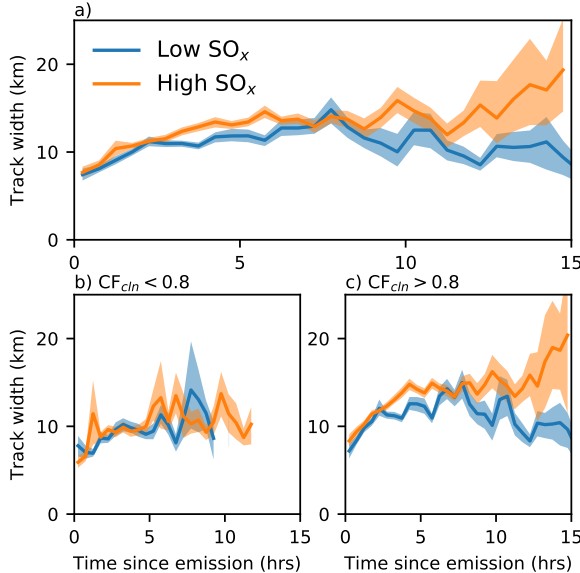

**Figure 5.** a) Shiptrack width as a function of time since emission for high (orange) and low (blue) $SO_x$ emitting ships. b) as (a) but only for segments with a $CF_{cln}$ <0.8, c) as (a) for segments with a $CF_{cln}$ <0.8. Averaged over half hour periods.

Similar to the low $CF_{cln}$ case, the gradual growth in the plume over time after this fast-growth period may be linked to cloud processes and mixing. Future model studies will be useful to identify the factors controlling the growth of the shiptrack width.

### 3.1.4 Controls on track length

Given the strong impact of cloud occurrence on the disappearence of shiptracks (Fig. 4), factors controlling liquid cloud
occurrence will also have an effect on the shiptrack length. The estimated inversion stability (EIS; Wood and Bretherton, 2006) has a strong link to low cloud cover in this region, but a comparison between high and low EIS environments shows only a 20% increase in the median lifetime, from five to six hours (Fig. 6a). This weak effect is primarily due to the prevalence of short shiptracks in this work and the stronger impact of meteorology on the lifetime of longer shiptracks.

Normalising by the total number of shiptracks remaining in the total sample highlights a much clearer role for meteorology,
with shiptracks in low average EIS environments only half as likely to have a lifetime longer than ten hours compared those with in a higher EIS environment, due to the lower cloud fraction (Fig. 6b). The low cloud fraction also impacts the sensitivity to $SO_x$ emissions, with the lifetime of shiptracks in low EIS environments being almost insensitive to $SO_x$ emissions. In contrast, the lifetime of high EIS shiptracks is much more sensitive to $SO_x$ emissions, as the high CF makes dissipation processes are more important. This is supported by studies aiming to produce a climatology of shiptracks finding more shiptracks in regions
of extensive low cloud cover (Schreier et al., 2007).



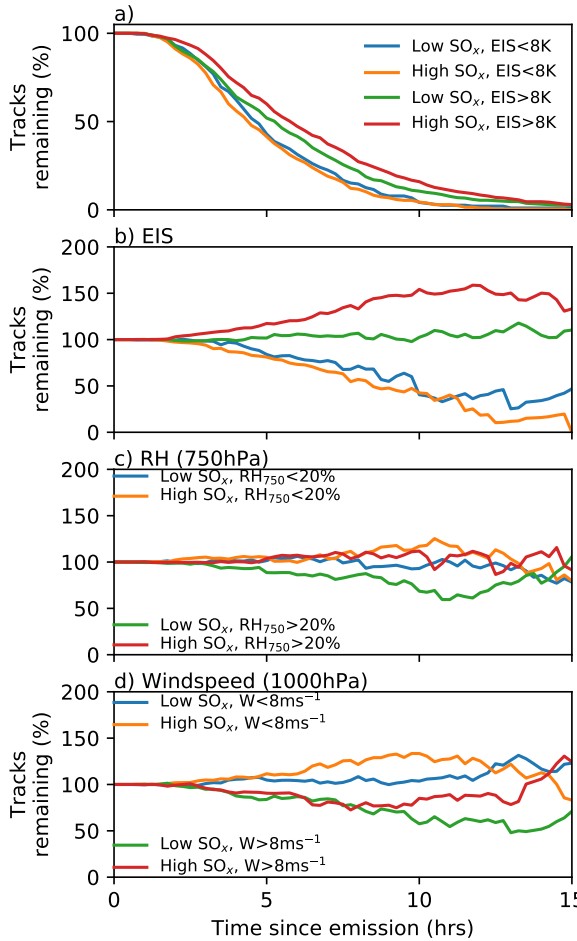

**Figure 6.** Shiptrack length as function of meteorological parameters. a) The percentage of tracks remaining for high and low $SO_x$ emissions for high and low EIS (averaged along the track). b) The percentage of tracks remaining as a fraction of the fraction of tracks remaining in the total dataset, comparing the impact of emissions at high and low EIS. c) As (b) but for cloud top relative humidity (750hPa). d) As (b) but for high and low 1000 hPa windspeed.

Despite a strong controlling influence on the formation of shiptracks (Gryspeerdt et al., 2019b), cloud top humidity has a much weaker impact on shiptrack length. Drier cloud tops typically indicate longer shiptracks, perhaps due to an increased in-cloud updraught and so higher sensitivity to aerosol (Lilly, 1968). A slight decrease in shiptrack length is observed for moister cloud tops.

5  Windspeed has a stronger influence on shiptrack length, with higher windspeeds producing shorter shiptracks. Although shiptrack formation is more common at high windspeeds (Gryspeerdt et al., 2019b), these shiptracks are typically shorter and have a length that is less sensitive to the size of the aerosol perturbation. This may be due to a number of factors. Sea salt production is enhanced at high windspeeds, which may lead to a decrease in $N_d$, perhaps through giant CCN production





(Lehahn et al., 2011; Gryspeerdt et al., 2016; McCoy et al., 2017), an increase in precipitation and so a reduction in the impact of the aerosol perturbation. An increase in surface fluxes at high windspeeds may also lead to a breakup of the cloud deck, resulting in shorter shiptracks.

### 3.1.5 Cloud fraction enhancement

As well as creating increases in $N_d$ that allow their detection, the shiptracks in this study also have a higher cloud fraction than the control region (Fig. 7). An average increase of around 3% over the first ten hours is observed, peaking around 3-4 hours after emission. The CF increase is not strongly correlated to the ship $SO_x$ emissions (Fig. 7a), but varies significantly as a function of the background cloud state. In relatively clean conditions ($N_{cln}$ <50cm$^{-3}$), the CF is increased by almost 10% within the first five hours after emission (Fig. 7b). In contrast, the CF in already polluted conditions does not increase significantly inside

the shiptrack (Fig. 7c), partly due to the higher CF found at higher $N_d$ (e.g. Gryspeerdt et al., 2016) restricting the opportunity for any further CF increase within the shiptrack.

  By estimating the ship aerosol trajectories, shiptrack are located even when there are no surrounding clouds (preventing the detection algorithm from operating). By selecting cases with a low $CF_{cln}$ (<10%), a 3-7% increase in cloud amount is found in the early stages of the shiptrack (Fig. 7d). This increase comes from an increase in CF in around 10% of segments within the

first 5 hours of the shiptrack (Fig. 7e). A CF increase is more common in cases with lower $SO_x$ increases, but does not appear to be correlated to the reanalysis background aerosol (Fig. 7f).

  Random fluctuations in the retrieved CF have a part to play in this apparent increase in cloud fraction. If the starting CF is zero, a fluctuation can only generate a CF increase, giving the appearance of an aerosol-limited CF. This is likely the case more than ten hours after emission, where the percentage of potential aerosol-limited cases stabilises at around 5%. As the impact of

random fluctuations is independent of the time since emission, this suggests that up to 5% of aerosol-limited cases in the early stages of the shiptrack are random fluctuations in the CF retrieval and that 5% of the clear-sky cases may be aerosol-limited (Fig. 7e).

  A 5% reduction in the clear sky cases with an increase in aerosol is a roughly 2% increase in the liquid cloud fraction in this region due to aerosol-limited cases. This is smaller than the total change in CF from potential $N_d$ variations found in Gryspeerdt

et al. (2016), making it a plausible measure for the fraction of aerosol-limited cases in this region. However, as it relies on gaps and dissipation of existing shiptracks, it is not clear whether this measure is representative of the wider region, particularly locations far from existing clouds.

### 3.2 Microphysics

  The previous section has shown that the background meteorological state has a controlling influence on the macrophysical

properties of shiptracks and their sensitivity to aerosol. These are not independent from the microphysical properties of the shiptrack, particularly the $\varepsilon_N$, which is closely related to shiptrack detection. Changes in the microphysical can also have a distinct impact on the potential radiative forcing from shiptracks. In this section, the changes in the $N_d$ and LWP are examined to investigate these controls in more depth and quantify the radiative response as a function of time.

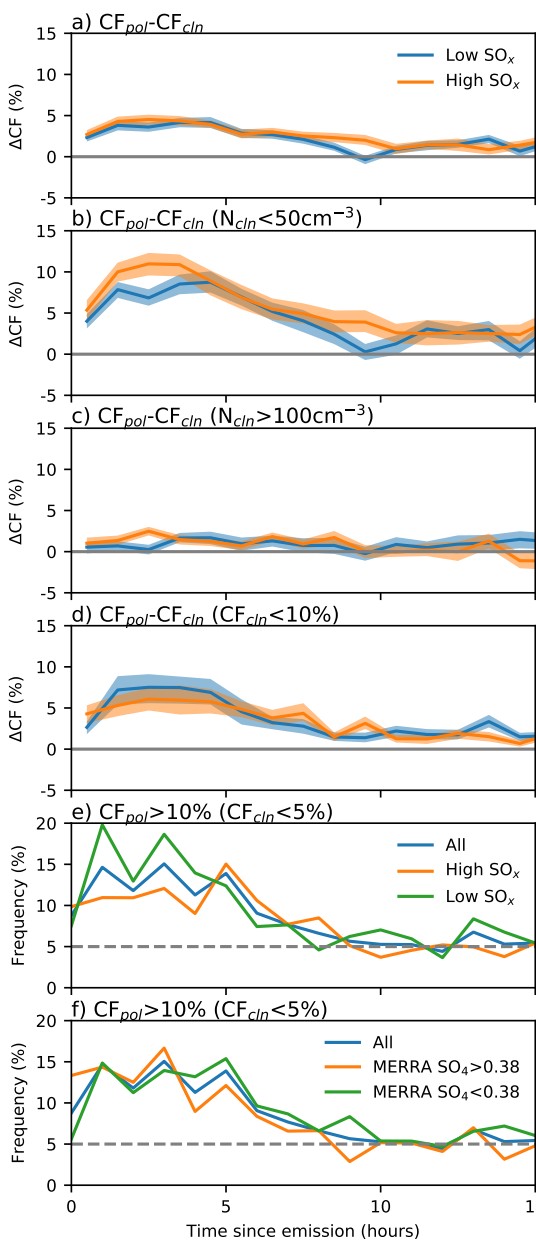

**Figure 7.** a) Cloud fraction in the polluted shiptrack regions compared to the corresponding clean locations. b) as (a) but only for cases where the $N_d$ in the clean region is $<50 \mathrm{cm}^{-3}$. c) as (a) but only for cases where the $N_d$ in the clean region is $>100 \mathrm{cm}^{-3}$. d) as (a) but only for cases where the $\mathrm{CF}_{cln}$ is $<10\%$. e) the percentage of cases with $\mathrm{CF}_{pol} >10\%$ and $\mathrm{CF}_{cln} <5\%$, stratified by the ship $\mathrm{SO}_x$ emissions. f) as (e) but stratified by reanalysis/background $\mathrm{SO}_4$. Grey lines are gridlines.





### 3.2.1 Polluted vs. detected

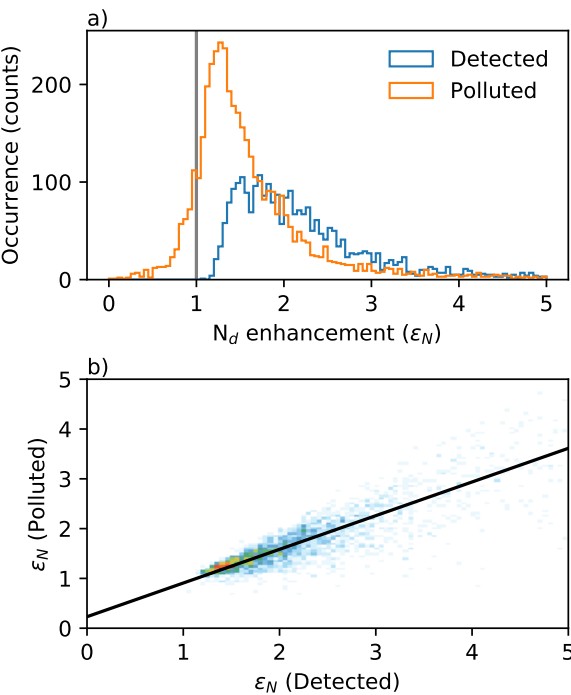

**Figure 8.** a) The $N_d$ enhancement ($\varepsilon_N$) for detected cases and all polluted cases two to three hours after emission for all cases with sufficient cloud. b) the relationship between polluted and detected for all cases in (a) where both metrics are derived.

Detected pixels (more than 2 standard deviations above the background $N_d$) are only a subset of the pixels within the shiptrack. In this section, the "polluted" $N_d$ (Fig. 1) is used as a measure of the $N_d$ within the shiptrack, enabling a consistent comparison between segments with and without a detected shiptrack. The detected $\varepsilon_N$ is positive by definition (Fig. 8a). As it includes (undetected) pixels with a lower $N_d$, the polluted $N_d$ is smaller than the detected $N_d$ and in some cases can lead to an $\varepsilon_N$ of less than one (typically in segments where there are no detected pixels).

In segments where both the polluted and detected $N_d$ exist, there is a close correlation between the polluted and detected $\varepsilon_N$ (r+0.9; Fig. 8b). This correlation rises further (to 0.95), when only segments with more than five detected pixels are included, indicating that the primary source of uncertainty is in the detected $\varepsilon_N$. This supports the use of the polluted $\varepsilon_N$ for characterising the shiptracks in this work.

### 3.2.2 $N_d$ development

Previous studies have shown a strong link between the ship emissions and the detected $\varepsilon_N$ (Gryspeerdt et al., 2019b). This enhancement peaks in the early stages of the shiptrack formation and decreases quickly with time (Fig. 9a). As it depends on





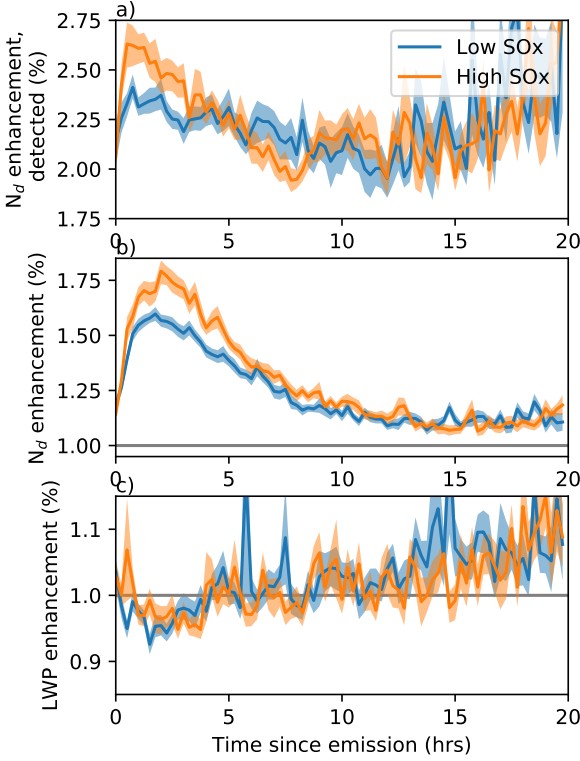

**Figure 9.** a) The change in $\varepsilon_N$ for detected pixels only as a function of time from emission, for low and high $SO_x$ emitting ships respectively. b), as (a), but including all the pixels within the shiptrack as the polluted pixels. c) As (b), but for the LWP instead of the $N_d$.

the relatively small number of detected pixels, the uncertainty in the enhancement is relatively high and the difference between the high and low $SO_x$ ships becomes small after five hours. Part of this uncertainty is due to a sampling effect, as weak $\varepsilon_N$ segments have no detected pixels and so are excluded from the calculation. This means that the average enhancement tends towards the lowest detectable enhancement.

5     A clearer signal is found using the polluted $\varepsilon_N$ (Fig. 9b). Following a quick increase in $\varepsilon_N$ during the shiptrack formation, a maximum is reached within about one hour (similar to the formation timescale in Fig. 3c). This timescale is longer than the formation timescale for the detected $\varepsilon_N$ (Fig. 9a), as the detected $\varepsilon_N$ only includes clouds where the ship $SO_x$ emissions have already had a detectable impact. The polluted $\varepsilon_N$ is almost 50% larger for ships with higher $SO_x$ emissions ($>0.13\,\mathrm{kg\,s^{-1}}$) than lower emissions, and a difference in shiptrack properties as a function of the initial aerosol perturbation is maintained
10    until almost 15 hours after emission. The decrease in $\varepsilon_N$ with time emphasises the importance of temporal development when considering aerosol impacts on clouds, especially these from an isolated aerosol source where there is no replenishment of the aerosol from an external source. This also demonstrates that if time since emission is not accounted for, the $\varepsilon_N$ (detected or polluted) is not a good measure of the aerosol perturbation in a shiptrack.





Although noisy, some patterns can be observed in the LWP enhancement ($\varepsilon_L = \frac{LWP_{pol}}{LWP_{cln}}$; Fig. 9c). There is an initial decrease of approximately 5% in the $\varepsilon_L$, followed by an increase in the $\varepsilon_L$ after two hours, becoming positive (increased LWP inside the shiptrack) at around seven hours since emission. There is no clear difference in $\varepsilon_L$ between ships with high and low $SO_x$ emissions.

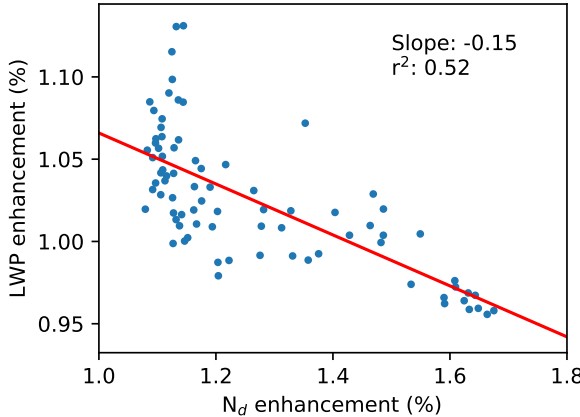

**Figure 10.** The relationship between the mean LWP and $N_d$ enhancements in Fig. 9. The slope and r$^2$ values for a linear regression are given in the plot.

The $\varepsilon_L$ will be addressed in later sections, but it is important to note that the temporal development in $\varepsilon_L$ and $\varepsilon_N$ will generate a relationship between them (Fig. 10). With $\varepsilon_N$ decreasing as $\varepsilon_L$ increases, this will create an apparent negative $N_d$-LWP sensitivity if the temporal development of the perturbation is not accounted for. This highlights the danger of using the $\varepsilon_N$ as a measure of the aerosol perturbation, particularly if the temporal development of the $N_d$ and LWP within the shiptrack is not taken into account.

### 3.2.3 Meteorology and $N_d$ development

As well as being a clear function of time, the $\varepsilon_N$ is also a function of meteorological state, such that even the $\varepsilon_N$ at a given time since emission is not necessarily a good measure of the aerosol perturbation. As the meteorological conditions can change along the length of a shiptrack, the results in the following sections are composited, with the meteorological conditions being calculated independently for each segment along the track. This allows situations where a track transects a region of variable meteorological conditions to be investigated.

The $\varepsilon_N$ is higher at low EIS (Fig. 11a), where it is also sensitive to ship $SO_x$ emissions with increased $SO_x$ enhancing the $\varepsilon_N$ further. However, the $\varepsilon_N$ for both the high and low $SO_x$ tracks becomes very similar after four hours. This is similar to the track length in Fig. 6b, which is also not a strong function of the aerosol perturbation. In both cases, the impact of meteorological variations dominates the variability from the magnitude of the aerosol perturbation. In contrast, while the $\varepsilon_N$ is





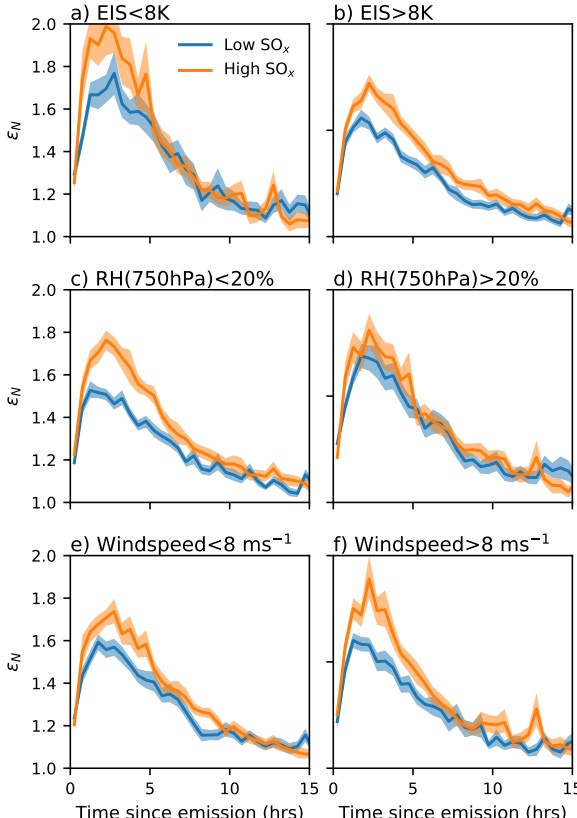

**Figure 11.** As Fig. 9b, but stratified by: a,b) estimated inversion strength (EIS), c,d) relative humidity at 750 hPa, e,f) 1000 hPa windspeed. Orange and blue are high and low $SO_x$ emissions respectively.

smaller (partially due to a larger $N_{cln}$; Bennartz and Rausch, 2017) in the high EIS cases (Fig. 11b), the signal from the aerosol perturbation remains to almost 20 hours (although the $\varepsilon_N$ itself is still a strong function of time).

Relative humidity also affects $\varepsilon_N$, with shiptracks at lower cloud top humidity being more sensitive to the initial aerosol perturbation (Fig. 11c). When the cloud top humidity is higher, the sensitivity of the $\varepsilon_N$ to the aerosol perturbation disappears
5 almost entirely (Fig. 11d). This is expected following previous work showing shiptracks are more likely to form in regions with a low cloud top humidity (Gryspeerdt et al., 2019b) and may be due to the higher in-cloud updraught expected with drier cloud tops (Lilly, 1968) enhancing the sensitivity to aerosol.

The dependence of the $\varepsilon_N$ development on surface windspeed may have a similar origin. The larger sensitivity of $\varepsilon_N$ to the aerosol perturbation at higher windspeeds (Fig. 11f) may be due to the increased surface fluxes promoting a larger in-cloud
10 updraught and hence a more aerosol-limited environment.



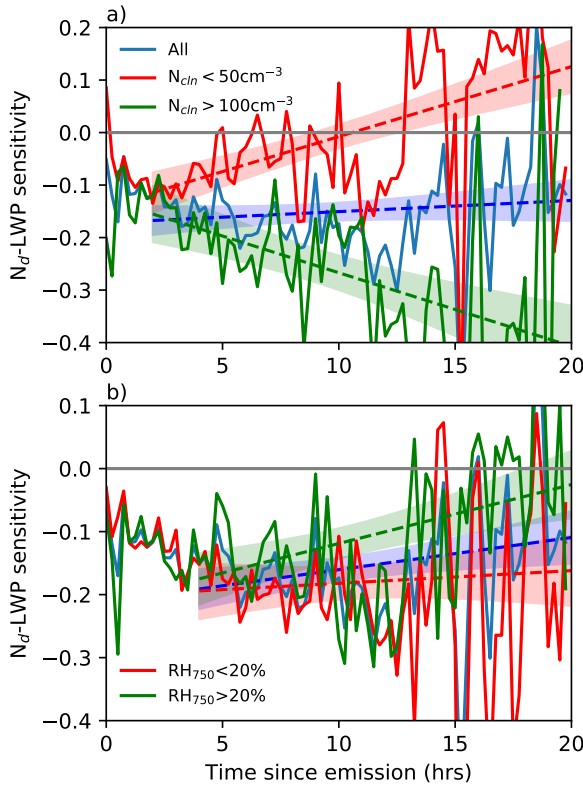

**Figure 12.** The LWP sensitivity to $N_d$ as a function of time since emission. a) shows the development of the sensitivity as a function of $N_{cln}$, b) as a function of RH(750hPa). The blue line is the same in both plots as a reference. Linear fits for the data from 2 to 10 hours after emission, along with a standard error on the fit are also shown. Note that this is calculated with the "detected" $\varepsilon_N$ to avoid uncertainties when dividing by small $\varepsilon_N$.

### 3.2.4 LWP sensitivity

The impact of aerosol on the liquid water path is often quantified as the sensitivity of LWP to $N_d$, $\frac{d\ln LWP}{d\ln N_d} = \frac{\ln \varepsilon_L}{\ln \varepsilon_N}$ (e.g. Han et al., 2002; Feingold, 2003). This sensitivity can be calculated directly from a shiptrack, comparing the polluted and corresponding clean regions, and has been used to quantify the strength of LWP adjustments to aerosol (e.g. Toll et al., 2019). However, as

5 demonstrated in the previous section, both $\varepsilon_N$ and $\varepsilon_L$ are functions of time since emission, such that a negative bias to the sensitivity will be induced if the time since emission is not accounted for (Fig. 10). Previous model and observation-based studies have suggested that the sensitivity is likely a function of the background cloud state (Gryspeerdt et al., 2019a), the large scale meteorology (e.g. Toll et al., 2019) and the time since emission (Glassmeier et al., 2020).

The average sensitivity (calculated individually for each segment using the $\varepsilon_L$ and polluted $\varepsilon_N$) is a clear function of time
10 from emission (Fig. 12a). Considering all the shiptracks together (blue line), the sensitivity almost instantaneously decreases





to -0.1, before slowly decreasing to -0.2 over the following 15 hours. Using $N_{cln}$, as a measure of the background cloud state (e.g. Gryspeerdt et al., 2019b), it is seen that not only is the sensitivity a function of time since emission, it is a strong function of $N_{cln}$ (Fig. 12a). A bi-directional LWP response has been observed in several previous studies (Han et al., 2002; Chen et al., 2012; Gryspeerdt et al., 2019a; Toll et al., 2019), hypothesised to be a combination of precipitation suppression (generating a

positive sensitivity of LWP to $N_d$) and aerosol dependent entrainment (generating a negative sensitivity Ackerman et al., 2004).

This bi-directional LWP sensitivity response is clear in (Fig. 12a). For both the clean and polluted background clouds, the sensitivity remains very similar until around two hours after emission. Here they diverge, with the sensitivity becoming more positive for a clean background and more negative for a polluted background. This is consistent with the precipitation suppression effect dominating in clean regions, but the enhanced entrainment being more important for polluted situations

(e.g. Gryspeerdt et al., 2019a). This behaviour continues to at least 20 hours since emission, although the sensitivities at long timescales should be treated with some caution due to the small number of shiptracks remaining at such long times. The exact values of the sensitivity at large times since emission also depend significantly on the choices made in the shiptrack pixel identification algorithm, but the humidity dependence of the results remains. At long timescales, the sensitivity would tend towards to the large scale statistics, which may be impacted by retrieval biases from correlated errors in the LWP and $N_d$

retrievals (Gryspeerdt et al., 2019a).

A similar (but smaller) divergence in the sensitivity is observed as a function of cloud top humidity (Fig. 12b). Both dry and moist cloud-top conditions maintain a similar sensitivity for the first four hours; at subsequent times, the drier cloud tops display a more negative sensitivity and the more humid cloud tops have a more positive sensitivity, with these conditions diverging increasingly over time. This is consistent with the impact of meteorology shown in previous studies (Ackerman et al., 2004;

Toll et al., 2019).

While these results support those found in previous studies, there are a number of important factors that must be taken into account when interpreting those studies. First, as demonstrated in model studies (Glassmeier et al., 2020), the $N_d$-LWP sensitivity is not constant with time. This introduces extra uncertainties into studies that do not account for this variation. Second, $\varepsilon_N$ has been shown to be a be a poor indicator of the aerosol perturbation under some meteorological conditions (Fig. 11). If

the $\varepsilon_N$ is primarily a factor of the local meteorology, this increases the potential impact of meteorological covariations on the sensitivity (e.g. Gryspeerdt et al., 2019a). Finally, the timescales involved may hint at the potential for a retrieval bias creating an extra negative sensitivity in these results (see Sec. 4.2).

### 3.2.5 LWP development

Averaging the changes in LWP over hour-long periods, the temporal changes in LWP after the aerosol emission become clearer

(Fig. 13). A decrease in LWP is observed at short timescales, followed by an increase in LWP at longer timescales (Wang and Feingold, 2009b). There is no clear difference in LWP evolution for the high or low $SO_x$ emissions.

As seen in the sensitivity, when looking at cases with a clean background (Fig. 13b), after a short term decrease in LWP, there is a strong increase in LWP after 5 hours since emission. This is likely due to precipitation suppression and the circulation





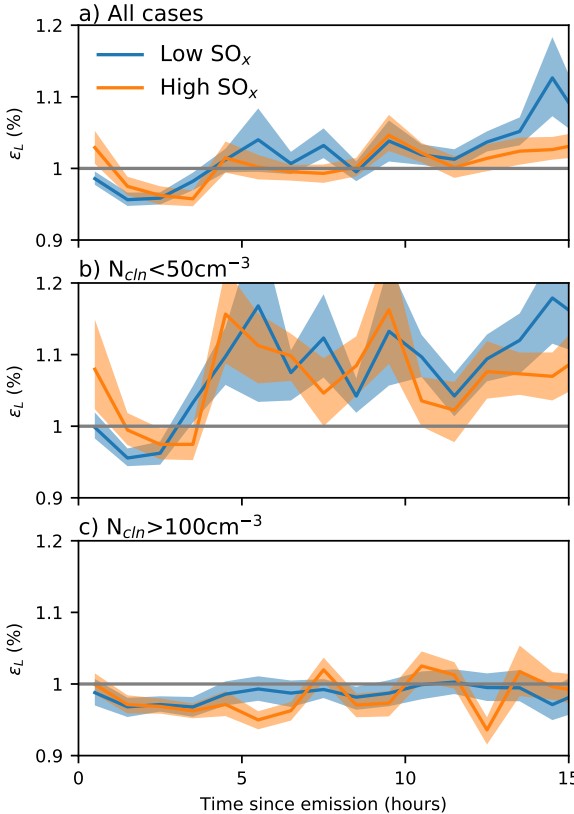

**Figure 13.** Development of LWP along shiptracks for all ships (a). b) clean background environments ($N_{cln}$ <50cm$^{-3}$). c) polluted background environments ($N_{cln}$ >100cm$^{-3}$).

adjustments created around a spatially limited aerosol perturbation (Scorer, 1987; Wang and Feingold, 2009b), particularly in an open-celled stratocumulus regime (Christensen and Stephens, 2011).

For shiptracks forming in polluted environments, there is a small decrease in LWP (Fig. 13c). This decrease is larger for shiptracks formed by larger SO$_x$ emissions, although the overall magnitude of the effect is smaller than the increase in cleaner conditions. Although the sensitivity appears to continue to increase, this appears to suggest that the increase in sensitivity is almost exactly offset by the decrease in $\varepsilon_N$. This is in contrast to the results from (Glassmeier et al., 2020), which suggests that the change in sensitivity comes from the LWP adjusting to the change in $N_d$. Further work is required to properly understand the extent to which shiptracks and the response to isolated pollution sources represent the actual response of cloud and particularly LWP to aerosol perturbations.





# 4 Discussion

## 4.1 Potential radiative forcing

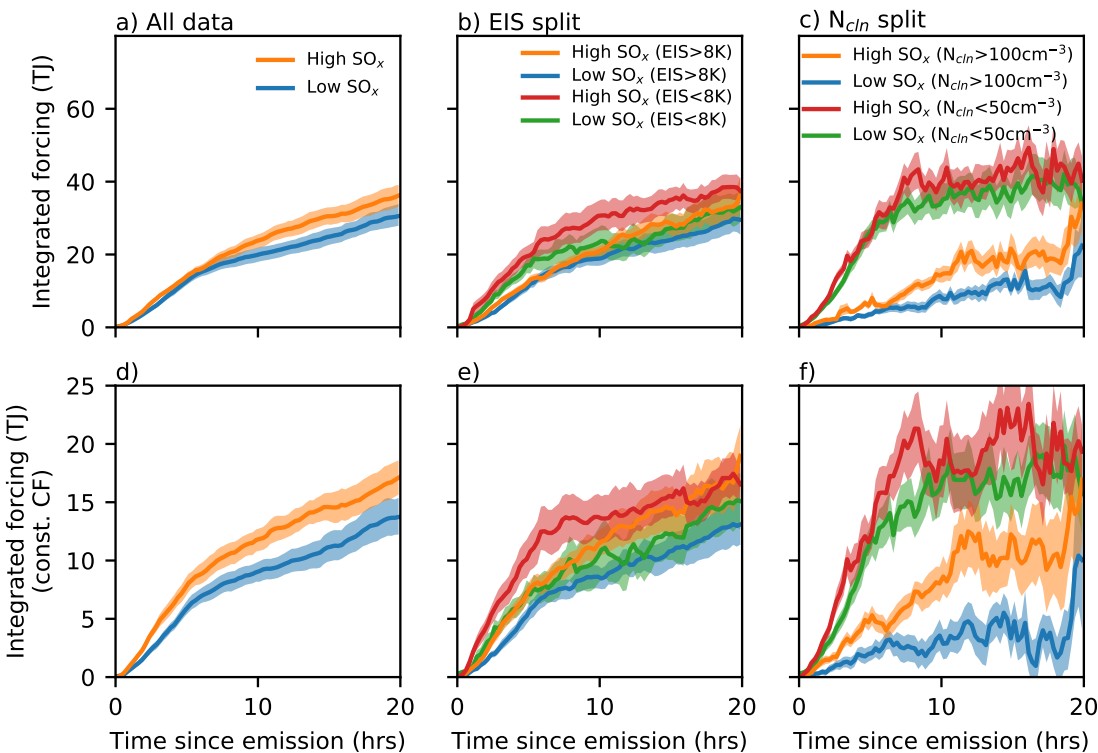

**Figure 14.** The integrated potential forcing as a function of time since emission including (top row) and excluding (bottom row) CF changes (Eq. (1)). Note the different scales. a,d) All shiptracks, separated by high and low $SO_x$ emissions. b,e) Separated by high and low $SO_x$ emissions for high EIS (>8K; orange and blue) and low EIS (<8K, red and green as high and low $SO_x$ respectively). c,f) As (b) and (d), but for high $N_{cln}$ (>100cm$^{-3}$; orange and blue) and low $N_{cln}$ (<50cm$^{-3}$; red and green).

The combination of these increases in $N_d$ (Fig. 11), LWP (Fig. 13) and CF (Fig. 7) produce an increase in reflected shortwave radiation, providing a way to compare the different perturbations with a similar metric. Integrating the potential radiative forcing along each shiptrack before creating a composite shows that about half the total radiative impact of the composite shiptrack comes in the first five hours (Fig. 14a). During this period, the $SO_x$ emissions of the ship do not have a strong impact on the integrated forcing. The $SO_x$ impact increases in the later stages of the shiptrack lifetime, where higher $SO_x$ emitting ships produce a larger $\varepsilon_N$ (Fig. 9b) later in their lifetimes. Excluding the CF change (Fig. 14d) shows that approximately half the total radiative effect of the shiptrack comes from CF increases. The weak correlation of these CF increases to the ship $SO_x$ emissions explains the insensitivity of the integrated forcing to $SO_x$ during the early stages of the track.



The liquid cloud fraction has previously been shown to be the primary control on shiptrack formation (Gryspeerdt et al., 2019b). This is strongly linked to the EIS, with more liquid cloud in more stable environments (Wood and Bretherton, 2006). However, despite a higher cloud fraction at high EIS, shiptracks formed in lower EIS environments have a larger radiative effect (Fig. 14b). This is partly due to the lower CF allowing the CF (and hence forcing) to increase within the shiptrack.

With a higher background CF, there is less scope for a forcing due to a CF increase (Goren et al., 2019). This interpretation is supported by the results at a constant cloud fraction (Fig. 14e), where after an initial difference in the forcing (due to the higher $\varepsilon_N$ at low EIS; Fig. 11), the integrated forcing for the high and low EIS populations is very similar after ten hours.

Although EIS has some control over the radiative effect of shiptracks, background $N_d$ have the largest impact on shiptrack radiative properties (Fig. 14c), with larger $\varepsilon_N$ (Gryspeerdt et al., 2019b), $\varepsilon_L$ (Fig. 13) and CF enhancement (Fig. 7) occurring

with a clean background. The forcing for the clean background ($N_{cln} < 50\mathrm{cm}^{-3}$) cases is mostly within the first seven hours, with very little increase in the integrated forcing after this time. This is likely due to the lower cloud fraction at a lower $N_{cln}$ (Gryspeerdt et al., 2016) both enhancing the forcing from CF increases in the early stages of the track and limiting the forcing from $\varepsilon_N$ increases in the later stages. In contrast, the integrated forcing from shiptracks in polluted backgrounds shows a relatively steady increase with time, although after twenty hours it still have less than half the integrated forcing of the

shiptracks in clean environments. These shiptrack locations were only estimated up to twenty hours from emission. Some very long-lived shiptracks have radiative impacts lasting several days (Goren and Rosenfeld, 2012), which would lead to significant changes at longer timescales.

## 4.2  Timescales

The $N_d$ response to the aerosol perturbation proceeds at a timescale approximating a boundary layer mixing timescale of

around half an hour (Figs. 3d, 9). The timescale for the precipitation suppression impact on LWP to become clear is around two hours (Fig. 12a). This increase in LWP is quicker than the four hour delay in Wang and Feingold (2009b) but slower than the almost instantaneous LWP response to a $N_d$ increase in Feingold et al. (2015). It may also indicate that the LWP response to precipitation/circulation effects is faster than model simulations suggest.

The aerosol impact through a modification of entrainment proceeds at a slower pace. In Fig. 12b, it takes four hours for

the impact of variations in cloud top humidity to begin to appear. As the impact of cloud top entrainment depends on the humidity, the timescale for humidity variations to impact the $N_d$-LWP sensitivity is the relevant timescale for the impact of aerosol-dependent entrainment. This compares well with the results from Glassmeier et al. (2020), which shows a characteristic timescale for the LWP adjustment via entrainment of 20 hours, which produces a 20% of the total change after four hours.

While these timescales match previous work, the $N_d$-LWP sensitivity adjusts very quickly immediately behind the ship

(Fig. 12). The sensitivity reaches -0.1 within the first 15 minutes, meaning that the LWP has changed on the same timescale as the $N_d$. While the $N_d$ can respond at this timescale (e.g. Wang and Feingold, 2009b), LWP adjustments depending on precipitation or entrainment appear to operate at longer timescales (Fig. 12). The almost instantaneous LWP adjustment may indicate a retrieval bias.





Both the LWP and $N_d$ are calculated using the cloud optical depth and the effective radius. As they both depend on the same retrieved variables, this leads to the potential for biases in the effective radius retrieval to lead to correlated errors in the $N_d$ and LWP, generating a negative sensitivity (e.g. Gryspeerdt et al., 2019a). As retrieval biases are insensitive to the time since emission, uncertainty in the effective radius retrieval could lead to this instantaneous negative sensitivity. One potential cause

of a retrieval bias could be in the droplet size distribution (DSD). The shape of the DSD affects the relative number of small and large droplets and hence the link between the $N_d$ and effective radius. Aircraft studies have shown a wider DSD in shiptracks compared to the surrounding cloud, which would create an uncertainty in retrieval of cloud properties in shiptracks (Noone et al., 2000). Future high spatial-resolution polarimeters may be able to resolve this ambiguity.

If the initial value of the sensitivity of -0.1 represents the impact of retrieval biases, this suggests the negative sensitivities

determined from previous studies of shiptracks should be smaller, resulting in a weaker LWP reduction in response to aerosol. Recent studies have suggested that the LWP adjustments inferred from shiptracks may be underestimated due to the lack of consideration of the shiptrack temporal development (Glassmeier et al., 2020). These model results are supported by the observational evidence presented in this work (Fig. 12), suggesting that the long term LWP sensitivity to $N_d$ may be larger than found in previous studies. It should be noted that the long-term temporal development of these shiptracks (particularly

those that contain no detected pixels), will tend to the large-scale statistics, which themselves may be subject to biases from correlated errors (e.g. Gryspeerdt et al., 2019a). However, these two factors suggest that the interpretation of current and future studies inferring LWP adjustments from natural experiments should consider the temporal development of the perturbation and the possibility of retrieval errors.

### 4.3   Future improvements

This work has shown that many shiptrack properties vary significantly along the length of the track. As shiptracks do not transmit information along their length, they can be considered a collection of semi-independent segments, with the same initial aerosol perturbation, but at different times since emission. Using the ship location and the local windfield for reanalysis, the time since emission is inferred, allowing the timescales of the relevant cloud and aerosol processes to be measured. However, the results in this work come with some caveats.

The MODIS images are still only a snapshot of the cloud field, such that the time axis determined from the ship and cloud motion is not a real time axis. The unperturbed clouds will also develop over the time period (Christensen et al., 2009; Kabatas et al., 2013), such that the "unperturbed" clouds here are not a true measure of the at the time of the aerosol perturbation, having evolved since then. The polluted clouds will also have evolved, not necessarily in the same manner. The climatological meteorological fields also affect both the clean and polluted clouds (15 hours can be several hundred kilometres). This generates

uncertainties that will be resolved in future work through the use of geostationary observations.

Although many shiptracks are intersected by CloudSat/CALIPSO, it is not enough to build up a picture of the precipitation development in these shiptracks. MODIS views every segment in each track, with 1,209 tracks, almost 100,000 segments are used in this study. CloudSat will typically only view one segment per track (if any), so that resolving the temporal to the same detail requires 80 times as many shiptracks. This will be investigated in more detail in a future study that will expand this





work to a global scale, enabling a more complete picture of the factors limiting and controlling aerosol perturbations on cloud properties.

Similarly, the results in this work are limited to a single year of data in a small region off the western coast of North America. While this is an important region for shiptrack formation (Schreier et al., 2007), it is unclear the extent to which these results

can be generalised globally. Further work is underway to expand the results presented here to establish their relevance across different cloud regimes.

## 5  Conclusions

By combining ship $SO_x$ emissions with satellite observations of shiptracks, previous studies have demonstrated the importance of the size of both the aerosol perturbation and the background cloud field for the strength and properties of shiptracks

(Gryspeerdt et al., 2019b). This work uses advected emissions locations to estimate the time since emission for locations along a shiptrack. This converts the distance along a shiptrack into a time-axis, providing a method for measuring the timescales of aerosol-cloud interactions from individual satellite images. This also provides an estimate for shiptrack locations in regions where they are too weak to be detected by existing methods (Fig. 2).

While ships with higher $SO_x$ emissions typically produce longer shiptracks, the median lifetime of the shiptracks studied in

this work is not a strong function of the ship emissions (Fig. 3). The role of ship emissions for shiptrack length varies by meteorological background, with shorter shiptracks being found in more unstable (higher EIS environments), where the shiptrack length is insensitive to the ship $SO_x$ emissions (Fig. 6b). In contrast, higher $SO_x$ emissions lead to a significant increase in track lifetime in more stable environments, due to the higher cloud fraction and stronger role of dissipation. Shiptracks disappear due to a lack of cloud and an insufficient $N_d$ perturbation roughly equally (Fig. 4). In a low EIS environment, the low cloud fraction

is the primary limit on shiptrack lifetime. At high EIS, the cloud fraction is typically large, leading to a greater sensitivity of shiptrack lifetime to the aerosol perturbation.

After an initial increase in track width (within the first five hours), track width is relatively insensitive to the time since emission (Fig. 5). In high CF environments (an indicator of closed cells/sea fog), the track width increases with $SO_x$ emissions. In low CF environments (indicating open cells), the track width is largely independent of $SO_x$ as the track width is largely

controlled by the cell width (Scorer, 1987).

The reanalysis windfield is used to locate shiptracks in otherwise cloud-free environments (Fig. 7). A significant increase in CF is found during the first 10 hours of the shiptrack lifetime, suggesting that around 5-10% of clear sky cases in this region are aerosol-limited. There is a large uncertainty on this number, due to potential random fluctuations in the cloud fraction retrievals. Further studies are required to establish the extent of these aerosol-limited cases across the global oceans.

The microphysical properties of the shiptrack, particularly the $N_d$ enhancement ($\varepsilon_N$), along with the enhancement in LWP are also shown to vary significantly with time since the aerosol emission (Fig. 9). The $\varepsilon_N$ quickly reaches a maximum around an hour after emission, before beginning to decrease slowly. The magnitude of the $\varepsilon_N$ maximum depends on the meteorological state, which also controls the sensitivity of the $\varepsilon_N$ to the $SO_x$ emissions. The cases with the largest $\varepsilon_N$ (such as at low EIS;





Fig. 11a), often have a lower sensitivity to the $SO_x$ emissions, in some cases showing no sensitivity to the size of the $SO_x$ perturbation (such as when the cloud top relative humidity is larger than 20%). The meteorological environment also affects how long the initial aerosol perturbation signal can be detected in the $\varepsilon_N$. In cases with a high EIS, large $SO_x$ emissions will have a larger $\varepsilon_N$ even at 15 hours since emission.

Although the $\varepsilon_N$ can remain correlated to the $SO_x$ perturbation, it is in general not a good measure of the aerosol perturbation. It is strongly dependent on time, decreasing with time since emission. The LWP enhancement ($\varepsilon_L$) tends to increase over time (Fig. 9). The temporal development of these properties generates an apparent negative $N_d$-LWP sensitivity if the time since emission is not take into account (Fig. 10). Even when accounting for the shiptrack development, the $N_d$-LWP sensitivity remains strongly dependent on the time since emission, along with the background cloud and meteorological state (Fig. 12).

This demonstrates that positive LWP adjustments are possible when the background cloud state is clean, with the impact becoming visible after about two hours. A four hour period is necessary before the cloud top humidity has an effect on the $N_d$-LWP sensitivity. A negative $N_d$-LWP sensitivity appears within 15 minutes of emission (Fig. 12). The speed of this adjustment hints that a retrieval error may be involved, as previous studies indicated that the LWP adjustment requires several hours to become visible (e.g. Feingold et al., 2015; Glassmeier et al., 2020). Correlated errors in the $N_d$ and LWP retrievals are one

possible explanation, which could potentially be caused by a change in the shape of the droplet size distribution (Noone et al., 2000), although in-situ studies are required to investigate this possibility. This temporal development of the $N_d$-LWP should be accounted for in studies of cloud adjustment to aerosol.

When combined, the radiative impact of the shiptracks investigated in this work is concentrated in the first 5-10 hours after emission (Fig. 14), with approximately half of the overall effect being due to the increase in CF in the early stages of the ship-

track lifetime. This emphasises the large potential role of cloud fraction adjustments to the overall potential radiative forcing (Goren and Rosenfeld, 2014). This strong impact of the CF increase means that shiptracks in lower EIS environments have a higher potential radiative forcing (Fig. 14b), although the background $N_d$ dominates, with shiptracks in clean environments producing a four times larger integrated radiative forcing over the 20 hour study period.

Although uncertainties remain, this study demonstrates how isolated aerosol perturbations can be used to measure the

timescales of aerosol impacts on cloud properties, showing that the $N_d$ perturbation is not often a good measure of the size of the aerosol perturbation and that meteorology and background cloud state have an important role in determining the sensitivity of cloud properties to aerosol. Together, the results in this work emphasise the importance of accounting for time when using observations of isolated aerosol perturbations to constrain aerosol-cloud interactions.

*Competing interests.* The authors have no competing interests.





*Acknowledgements.* Ship emissions were derived using location data from ExactEarth. EG was supported by a Royal Society University Research Fellowship (URF/R1/191602). TG was funded by the European Union's Horizon 2020 Research and Innovation Programme under Grant Agreement 821205 (FORCeS)



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
