# Peer review of "Observing the timescales of aerosol-cloud interactions in snapshot satellite images"

_Atmospheric Chemistry and Physics, 2020_

## Referee Comment (RC1) · Anonymous Referee #1 · 29 Oct 2020

Overall comment

This study makes several key contributions to our understanding of the aerosol indirect effect using ship tracks. It goes beyond classical approaches to discover that cloud responses are time dependent in ship tracks. Several conclusions are drawn from this study which will lead to advancements in the fields of satellite remote sensing and atmospheric modeling via the newly identified satellite retrieval biases in ship tracks and LWP timescale constraints. My one concern is based on the use of cloud retrievals (Nd) to detect polluted pixels comprising ship tracks. It's not clear but the use of this data appears to potentially result in fewer detected polluted pixels compared to using classical near-infrared channel data (as in Segrin et al. 2007, https://doi.org/10.1175/2007JAS2308.1). The use of Nd data for detection and analysis may otherwise influence the interpretation of the results. Other than that, the paper is great and I have speckled in just a few other minor points.

Minor comments Pg3 :23 Please define "Fretchet distance" with either a reference or short description.

Pg3: 24 1,209 ship tracks are left but out of how many did you begin with? Are the decreases in this number related to the same factors outlined in figure 4?

Pg3: 25 Note, Segrin/Christensen methods use near infrared radiances to detect polluted pixels along ship tracks but figure 1 uses Nd.

Pg3: 30: "groups far from the track central location are classed as.." what does "far" mean in this context? This seems to be the method to remove nearby adjacent tracks but, how far is too far in this context? What if the track itself is very wide (up to 50 km) with some reflective edges as can happen especially in open cell cloud regimes. What is the widest tracks the algorithm can detect? I'm asking because if it fails to pick up such wide tracks in general this may offer an explanation for why the width tends to asymptote in figure 5 at great distances.

Pg5: L5-6: Based on this analysis it appears that the Nd retrieval is used to detect ship tracks since there is a lack of retrievals in the white background case considered in Fig. 2b. This would not be the case if raw near infrared reflectances were used instead for track detection. I am concerned that part of the conclusions drawn from this paper are dependent on missing retrieval data from MODIS when simply using the raw reflectances might be sufficient to detect these "missing Nd" pixels.

How is cloud fraction calculated? From the satellite retrieved Nd or from the standard collection 6 product? If it is calculated from Nd retrievals this may be problematic since it can be missing due to a variety of retrieval reasons (e.g. sun-glint) and it is used for the track detection.

What fraction of tracks are found by using the reanalysis winds where coherency in the

track is lost?

Pg8: L5: "these?"

Pg8: L5-12: Durkee et al. (2000), https://doi.org/10.1175/1520-0469(2000)057<2542:CSTC>2.0.CO;2 found the average time is about 20-25 mins from which the aeorsols are emitted to intercepting the cloud layer. This is also much shorter than the average reported here. It is, thus, surprising that it can sometimes take as long as 3-4 hours (the average lifetime of a ship track) before the creation of the track. How does the author know that these tracks are not created by other nearby ships? Has this been visually verified for some of the cases? When the pollution is dispersed and form tracks much later is the track wider on average as one might expect due to dispersion? Presumably, these cases occur when the cloud layer is decoupled from the surface. Liu et al. (2000), https://doi.org/10.1175/1520-0469(2000)057<2779:MOSETA>2.0.CO;2 suggests that when the boundary layer is decoupled the emissions from ships may take considerably longer to reach cloud base. This is nonetheless a very interesting result and nicely depicts there is more variability in the 20-25 mins originally demonstrated by Durkee et al. (2000). It may be worth mentioning more prominently in the text abstract or elsewhere.

Pg9: 8. But aren't cloud retrieved Nd required to detect ship tracks in the first place? Wouldn't this be improved if near IR data was used instead?

Pg22: 14: "it still have less" check grammar.

Pg22: 30: One of the leading order terms in the cloud retrieval bias is the assumption on the cloud drop size distribution spectrum (Painemal and Zuidema, 2011, https://doi.org/10.1029/2011JD016155) and drizzle. Perhaps, the DSD is incorrectly assumed for ship tracks and this would offer, as you suggest, an explanation for the LWP bias close to the head of the track. I wonder, however, if there is a physical argument for decreases in LWP too? A considerable amount of latent heat is released by the rapid condensation of cloud droplets when the plume first mixes with the cloud?

Enhanced buoyancy production could lead to stronger entrainment, downdrafts and evaporation too (Cotton et al. 1995, Earth Science Reviews 39, 169–206.).

Pg23: l33: it took until near the end of the manuscript to find out the number of samples in this study this information should come much sooner – maybe add N_samples in the method section.

---

## Referee Comment (RC2) · Anonymous Referee #2 · 7 Nov 2020

General comments:

This study proposes a method that uses snapshot polar-orbit satellite measurements, together with meteorological reanalysis data, to analyze time dependence of aerosol-cloud interactions for shiptrack phenomena. The authors conduct a careful analysis for identifying satellite pixels influenced by ship-emitted pollutants and associated cloud properties (Nd, LWP and cloud fraction), which are then investigated as a function of time since emission inferred from the distance from the source ship and near-surface wind fields. As a consequence of such an analysis, the authors obtained a series of interesting results regarding temporal evolutions of cloud responses to aerosol perturbations. I think this is a very nice study proposing a novel methodology that adds "time dimension" to the aerosol-cloud interaction analysis, and the results obtained are also

insightful for understanding the time-dependent processes of aerosol impacts on cloud. I would recommend this paper be published in Atmospheric Chemistry and Physics after my specific questions/concerns listed below are addressed appropriately.

Specific comments:

Page 3, Line 29: "...more than 2 standard deviations above the background" What variable/parameter do the authors talk about for this standard deviations criteria?

Page 5, Line 6: "these shiptracks are typically shorter and have a length that is less sensitive to the size of the aerosol perturbation" Is this argument based on Figure 6d? I see some differences of the shiptrack length between low and high SOx emissions in the plot. Where in the plot do the author refer to for "less sensitive to the size of the aerosol perturbation"?

Page 12, Line 14: "This increase comes from an increase in CF in around 10% of segments within the first 5 hours of the shiptrack (Fig. 7e)" I don't understand this statement. Can the authors clarify how Fig. 7e is interpreted to reach this statement?

Page, Line 17-22: I don't understand this whole paragraph. Can the authors instruct how several statements contained in the paragraph are derived from specific characteristics of Fig. 7e? I could not follow the argument just looking at Fig. 7e. I would appreciate the authors' guide on where to look at in Fig. 7e for each statement in this paragraph.

Page 12, Line 25: "making it a plausible measure for the fraction of aerosol-limited cases in this region" Probably because of my lack of understanding for the previous paragraph, I don't understand how this statement is derived. Can the authors explain it?

Page 15, Line 10: "until almost 15 hours after emission" In Fig. 9b, I don't see the difference between the low and high emission cases for 15 hours – I see the difference until about 10 hours. Can you clarify what "15 hours" refers to?

[Figure]

Page 16, Line 6: "With eN decreasing as eL increases" What is a physical cause for this anti-correlation between eN and eL?

Page 17, Line 5: "shiptracks are more likely to form in regions with a low cloud top humidity" Can the authors briefly describe a possible mechanism for this?

Page 20, Line 5: "this appears to suggest that the increase in sensitivity is almost exactly offset by the decrease in eN" How is this statement derived from? Which figure should the reader refer to?

Figure 14: This is a very useful plot, and I'm also curious how time series of potential radiative forcing (PRF) itself looks like. Can the authors also show them, which should be time derivatives of the integrated forcings shown here?

Page 22, Line 32: "The almost instantaneous LWP adjustment may indicate a retrieval bias" Can the authors briefly discuss how instantaneous negative response of LWP arises from retrieval errors? I don't understand why the DSD-relevant retrieval bias is a potential cause for the negative sensitivity of LWP although discussed in Page 23, Line 4.

Page 26, Line 27: "around 5-10% of clear sky cases in this region are aerosol-limited" Where does this conclusion come from?

Minor points:

Page 3, Line 5: properties (Nd, LWP) shiptracks -> properties (Nd, LWP) of shiptracks

Page 3, Line 26: based the method -> based on the method

Page 8, Line 5: These -> There

Page 10, Line 13: Delete "are"

Page 12, Line 31: microphysical -> microphysics

Page 19, Line 24: Delete "a be" prior to "a poor"

Page 24, Line 4: the extent to which -> to which extent?

Page 25, Line 8: take -> taken

---

## Author Comment (AC1) · 3 Feb 2021

**Response to Reviewer 1**

This study makes several key contributions to our understanding of the aerosol in-direct effect using ship tracks. It goes beyond classical approaches to discover that cloud responses are time dependent in ship tracks. Several conclusions are drawn from this study which will lead to advancements in the fields of satellite remote sensing and atmospheric modeling via the newly identified satellite retrieval biases in ship tracks and LWP timescale constraints.
* * *
*: My one concern is based on the use of cloud retrievals (Nd) to detect polluted pixels comprising ship tracks. It's not clear but the use of this data appears to potentially result in fewer detected polluted pixels compared to using classical near-infrared channel data (as in Segrin et al. 2007). The use of Nd data for detection and analysis may otherwise influence the interpretation of the results. Other than that, the paper is great and I have speckled in just a few other minor points.*

**Reply**: We thank the reviewer for their useful comments, we address their points in turn below. We note that some of the results in the figures are slightly modified due to a change in the shiptrack detection function for very wide shiptracks, although this doesn't impact the conclusions. Some text changes have been made to improve readability and Fig.s 10 and 11 have been reordered. Line numbers refer to the diffed version

We start with the key question about the identification of shiptracks in this work compared to previous studies. The reviewer raises an important point, which was not adequately explained. A new section has been added in the discussion about the shiptrack sampling and the potential impact in can have on the results (P25L13). We also explain the key points here.

As noted, previous work relied on near-IR data (typically 3.7um) to identify shiptracks. These wavelengths is primarily sensitive to effective radius variations, which enable an easy identification of the shiptrack following the Twomey effect. However, not all shiptracks are easily visible in the near-IR imagery. The example in Fig. R1 shows a case (A), where a shiptrack visible in the $N_d$ field is not visible in near-IR imagery. Similarly, a shiptrack (B) that is initially visible in the near-IR imagery apparently disappears, while remaining visible in the $N_d$ imagery.

This improved contrast and detectability in cloud covered scenes was the primary reason for using the $N_d$. The second reason is that relying on the near-IR (and hence $r_e$) biases shiptrack detection away from shiptracks with a LWP increase. If the liquid water path increases inside a shiptrack, this would increase the $r_e$, potentially to an extent that the track itself is no longer visible in near-IR imagery. It would remain visible in $N_d$, due to an increased optical depth. Using the near-IR channel only for identifying shiptrack thus depends on a decrease in $r_e$, which makes a LWP reduction more likely. This may lead to a negative bias in the LWP sensitivity, as it is unable to detect cases where the LWP increases.

There is a tradeoff - as the reviewer mentions, the $N_d$ is only retrieved

[Figure]

Figure R1: A section from the MODIS Aqua 2015.001.2210 Granule, showing shiptracks in the Californian deck. a) The $3.7\,\mu$m reflectance, b) the adiabatic $N_d$. The colourscales have been adjusted to give roughly equal contrast across the image.

in fully-cloud pixels. This means that some shiptracks that would have been detected by the near-IR channel but where no $N_d$ is retrieved, are excluded from this work. As this sampling bias has less of a direct effect on the $N_d$-LWP sensitivity (where the primary effect on the forcing is in high CF locations), we feel that it is an appropriate sampling method for this work. For future studies, we are looking at the most appropriate way to combine these two data sources to "fill the gaps" in the $N_d$ retrieval and build a more complete library of shiptracks.

We also note slight variations in defining the shiptrack edge can also have an impact on the results in this work. If the shiptrack itself is not excluded from the background $N_d$ variance calculation (when identifying polluted pixels), the detected polluted pixels are located closer to the centre of the shiptrack. This can lead to different values for the $N_d$-LWP sensitivity. Providing a strong constraint on the value of the $N_d$-LWP sensitivity from shiptracks will require a better understanding of these sampling and methodological effects.

**Minor comments**
* * *
***Pg3 :23****: Please define "Fretchet distance" with either a reference or short description.*
**Reply**: A short description has been added
* * *
***Pg3:24****: 1,209 ship tracks are left but out of how many did you begin with? Are the decreases in this number related to the same factors outlined in figure 4?*
**Reply**: 2,896 shiptracks are identified in this region during 2015, but not all of them could be linked to the generating ship. This difference is due to an exogenous factor (the difficulty of accessing shipping data), rather than a systematic effect as a function of the cloud properties, so is unlikely to bias the results in this work.
* * *
***Pg3: 25****: Note, Segrin/Christensen methods use near infrared radiances to detect polluted pixels along ship tracks but figure 1 uses Nd.*
**Reply**: This is now noted in the paragraph (and covered in the discussion - P25L13).
* * *
***Pg3: 30:****: "groups far from the track central location are classed as.." what does "far"mean in this context? This seems to be the method to remove nearby adjacent tracks but, how far is too far in this context? What if the track itself is very wide (up to 50 km) with some reflective edges as can happen especially in open cell cloud regimes. What is the widest tracks the algorithm can detect? I'm asking because if it fails to pick up such wide tracks in general this may offer an explanation for why the width tends toasymptote in figure 5 at great distances.*
**Reply**: That is a good point. The current method would fail to detect the whole track if the high $N_d$ pixels were widely spaced across the track (with

many low $N_d$ pixels inbetween). However, wide tracks on their own are not enough to cause the algorithm to fail and around 20% of the identified segments have a track wider than 20 km. Determining improved methods for shiptrack identification would be an interesting topic for future work. Neural net methods (e.g. Yuan et al., 2019), would be an interesting route.

Following this comment, we investigated a potential impact on the track width in the track detection function. To restrict the clean region to near that track, only pixels within 20 pixels of the track centre are considered. This has the effect of limiting the maximum track width around 40-50 km. We increased this limit to 50 km (maximum width of 100-120 km. This boosts the track widths slightly, leading to a slightly more linear increase in track width with time. The only other plots affected by this modification are the integrated potential radiative forcing (which increases with the slightly increased track width) and the LWP-$N_d$ sensitivity plot (Fig. 12b), where the impact of cloud top relative humidity is reduced (although maintains the same sign). These plots have been updated in the revised version of the paper.
* * *
**Pg5: L5-6::** *Based on this analysis it appears that the Nd retrieval is used to detectship tracks since there is a lack of retrievals in the white background case considered in Fig. 2b. This would not be the case if raw near infrared reflectances were used instead for track detection. I am concerned that part of the conclusions drawn from this paper are dependent on missing retrieval data from MODIS when simply using the raw reflectances might be sufficient to detect these "missing Nd" pixels.*

**Reply**: As noted above, this is a good point and we have included a section in the discussion to address it. Using the raw reflectances also misses some shiptracks and is not a perfect solution. On balance, we believe that the advantages of using $N_d$ in this work outweigh the potential issues.
* * *
*: How is cloud fraction calculated? From the satellite retrieved Nd or from the standard collection 6 product? If it is calculated from Nd retrievals this may be problematic since it can be missing due to a variety of retrieval reasons (e.g. sun-glint) and it is used for the track detection.*

**Reply**: The cloud fraction is determined from the "Cloud_Phase_Optical_Properties" variable, now noted in methods section. This is less than the cloud mask cloud fraction, but as noted, greater than the fraction of valid $N_d$ retrievals (the $N_d$ retrievals are further filtered following the criteria in (Grosvenor et al., 2018)). If this comment is directed towards Eq. 1, we do not believe the using $N_d$ for locating the track is a significant issue, as the track can still be located (using the emissions trajectory) even if there is no $N_d$ data. This means that a CF can always be calculated. Sun-glint effects would increase uncertainties, but are unlikely to produce a systematic bias in the properties of the shiptracks compared to their environment, as glint varies over a much larger spatial scale.
* * *
**Fig. 7?:** *What fraction of tracks are found by using the reanalysis winds where coherency in the track is lost?*

**Reply**: Assuming this is asking for shiptracks detected beyond the of the hand-identified track, 35% of all detected segments are found outside the hand identified region (R2 in Fig. 1). As only 24% of segments are in R2, they are much more likely to be detected as shiptracks. 61% of segments in R2 have a detected shiptrack, compared to 8% of segments in R4 (beyond the end of the hand-identified shiptrack.
* * *
*Pg8: L5:*: "these?"
**Reply**: Amended, thanks
* * *
*Pg8:L5-12:*: *Durkee et al.(2000) found the average time is about 20-25mins from which the aerosols are emitted to intercepting the cloud layer. This is also much shorter than the average reported here. It is, thus, surprising that it can sometimes take as long as 3-4 hours (the average lifetime of a ship track) before the creation of the track. How does the author know that these tracks are not created by other nearby ships? Has this been visually verified for some of the cases? When the pollution is dispersed and form tracks much later is the track wider on average as one might expect due to dispersion? Presumably, these cases occur when the cloudlayer is decoupled from the surface. Liu et al. (2000) suggests that when the boundary layer is decoupled the emissions from ships may take considerably longer to reach cloud base. This is nonetheless a very interesting result and nicely depicts there is more variability in the 20-25 mins originally demonstrated by Durkee et al. (2000). It may be worth mentioning more prominently in the text abstract or elsewhere.*
**Reply**: The shiptracks and trajectories used in this work are all based on hand identification, aided by the emission trajectory estimates. For the cases where the initial identification is many hours from the ship, this is often a case without a defined shiptrack "head", but where the polluted region of the cloud can be unambiguously linked to the ship (through the emissions trajectories). In cases where shiptracks intersect, or due to ships missing from our database, there is always the possibility of shiptracks being assigned to the wrong ship, but we believe these cases to be rare.

However, this does highlight a potential ambiguity around the time since detection. We have clarified in this section (P7L15) that the time to initial observation is not necessarily the time to the initial shiptrack formation. In cases without a detectable shiptrack head, the time to observation will be longer than the time to initial shiptrack formation. We don't yet have a good measure of the importance of these non-head cases, but aim to build a better climatology with future geostationary studies.
* * *
*Pg9: 8.*: *But aren't cloud retrieved $N_d$ required to detect ship tracks in the first place? Wouldn't this be improved if near IR data was used instead?*
**Reply**: As mentioned above, this may have some effect on the detection of very thin tracks. However, the tradeoff is that weaker tracks in cloud-covered regions can be detected.
* * *
***Pg22: 14:***: *"it still have less" check grammar.*
**Reply**: Amended
* * *
***Pg22: 30***: *One of the leading order terms in the cloud retrieval bias is the assumption on the cloud drop size distribution spectrum (Painemal and Zuidema, 2011,https://doi.org/10.1029/2011JD016155) and drizzle. Perhaps, the DSD is incorrectly assumed for ship tracks and this would offer, as you suggest, an explanation for the LWP bias close to the head of the track. I wonder, however, if there is a physical argument for decreases in LWP too? A considerable amount of latent heat is released by the rapid condensation of cloud droplets when the plume first mixes with the cloud? Enhanced buoyancy production could lead to stronger entrainment, downdrafts and evaporation too (Cotton et al. 1995, Earth Science Reviews 39, 169-206.).*
**Reply**: Many thanks for these references and suggestions. We are not clear on the cause of the near-instantaneous LWP adjustment and other potential mechanisms are welcome. We had considered entrainment as a potential cause for the decrease in observed LWP, but as cloud top entrainment would take time to reduce the LWP. For the fast response of LWP, it seems likely that the apparent change in LWP has to be directly related to the droplet size spectrum. As the reviewer notes, an increase in droplet surface area in a polluted cloud may lead to an increased condensation rate. However, this would lead to an increase in LWP and so seems unlikely to explain the results here. We have included a brief note on this pathway in the manuscript (P24L1).
* * *
***Pg23: l33***: *it took until near the end of the manuscript to find out the number of samples in this study this information should come much sooner - maybe add N_samples in the method section.*
**Reply**: The number of segments has been added to the method section where the segments are introduced (P6L1)

**Response to Reviewer 2**

General comments:
* * *
**This study proposes a method that uses snapshot polar-orbit satellite measurements,together with meteorological reanalysis data, to analyze time dependence of aerosol-cloud interactions for shiptrack phenomena. The authors conduct a careful analysis foridentifying satellite pixels influenced by ship-emitted pollutants and associated cloud properties (Nd, LWP and cloud fraction), which are then investigated as a function oftime since emission inferred from the distance from the source ship and near-surfacewind fields. As a consequence of such an analysis, the authors obtained a series ofinteresting results regarding temporal evolutions of cloud responses to aerosol perturbations. I think this is a very nice study proposing a novel methodology that adds "time dimension" to the aerosol-cloud interaction analysis, and**

*the results obtained are also insightful for understanding the time-dependent processes of aerosol impacts on cloud. I would recommend this paper be published in Atmospheric Chemistry and Physics after my specific questions/concerns listed below are addressed appropriately.:*

**Reply**: We thank the reviewer for their useful comments, which are addressed below. We note that some of the results in the figures are slightly modified due to a change in the shiptrack detection function for very wide shiptracks (see response to reviewer 1), although this doesn't impact the conclusions. Some text changes have been made to improve readability and Figs. 10 and 11 have been reordered. Line numbers refer to the diffed version.

**Specific comments:**
* * *
**Page 3, Line 29::** *"...more than 2 standard deviations above the background" What variable/parameter do the authors talk about for this standard deviations criteria?*

**Reply**: $N_d$ is used to identify the shiptracks, this is now noted in the paragraph
* * *
**Page 5, Line 6::** *"these shiptracks are typically shorter and have a length that is less sensitive to the size of the aerosol perturbation" Is this argument based on Figure 6d? I see some differences of the shiptrack length between low and high SOx emissions in the plot. Where in the plot do the author refer to for "less sensitive to the size of the aerosol perturbation"?*

**Reply**: Given the text, does this comment refer to Page 11, line 6? In that case, yes, this is intended to refer to Fig. 6d. As the reviewer notes, there is a difference in shiptrack length between the high and low $SO_x$ cases. This difference appears earlier in the low windspeed case, which is why it was suggested that these high windspeed cases were less sensitive to the aerosol perturbation. This has been modified to "initially less sensitive" (P12L14)
* * *
**Page 12, Line 14::** *"This increase comes from an increase in CF in around 10% of segments within the first 5 hours of the shiptrack (Fig. 7e)". I don't understand this statement. Can the authors clarify how Fig. 7e is interpreted to reach this statement?*

**Reply**: The sentence was confusing. It was intended to highlight that although the $\Delta$CF in Fig. 7d is around 5%, for the cases where $CF_cln$ is less than 5%, only 10% of segments have a $CF_{pol} > 10\%$. This suggests that the observed increased in CF is coming from a small number of shiptracks that have a large $\Delta$CF, rather than a large number of tracks with only a small CF increase. The sentence has been re-worded to make this clearer (P12L30).
* * *
**Page 12, Line 17-22::** *I don't understand this whole paragraph. Can the authors instruct how several statements contained in the paragraph are derived from specific characteristics of Fig. 7e? I could not follow the argument just looking at Fig. 7e. I would appreciate the authors' guide on where to look at*

*in Fig. 7e for each statement in this paragraph.*
**Reply**: This paragraph was aiming to highlight the importance of random errors in the cloud fraction retrieval in generating an apparent aerosol-limited regime. By selecting cases with a $CF_{cln}$ close to zero, if the corresponding $CF_{pol}$ is zero, a small random error in $CF_{pol}$ could only generate a positive $\Delta CF$, increasing the fraction of apparently aerosol-limited cases. As this bias is time-independent, we take the asymptote of Fig. 7e as an estimate of this effect (5% of segments). Given the apparent frequency of aerosol-limited cases is over 10% within 5 hours since emission, this suggests that around 5% of clear-sky cases in this region are clear due to a lack of available CCN. We appreciate that this is a rather hand-wavy estimate of a potentially complex bias in the cloud fraction retrieval, but with the caveats included, we feel it is still a useful and important result to present. The paragraph has been re-written to make it clearer.
* * *
***Page 12, Line 25::*** *"making it a plausible measure for the fraction of aerosol-limited cases in this region" Probably because of my lack of understanding for the previous paragraph, I don't understand how this statement is derived. Can the authors explain it?*
**Reply**: This paragraph has been largely removed and the remains merged with the previous paragraph. The comparison to previous work was not entirely helpful as a comparison for aerosol limited cases. The caveat about how this estimate is derived has been kept, given its importance for interpreting the result (P14L12).
* * *
***Page 15, Line 10::*** *"until almost 15 hours after emission" In Fig. 9b, I don't see the difference between the low and high emission cases for 15 hours - I see the difference until about 10 hours. Can you clarify what "15 hours" refers to?*
**Reply**: This depends on the smoothing does to the curve in Fig. 9b., but as noted, the difference is not clear at 15 hours. This has been modified to "more than 10 hours" (P15L13).
* * *
***Page 16, Line 6::*** *"With eN decreasing as eL increases" What is a physical cause for this anti-correlation between eN and eL?*
**Reply**: It is not clear that there should be a physical relationship between them, at last not a causal one. $\epsilon_N$ might be expected to decrease over time due to dilution (and precipitation in some cases), from an initial large value. $\epsilon_L$ in contrast might take time to change (with either an increase or a decrease), depending on time-sensitive processes such as cloud top mixing and precipitation. The different timescales for the processes would give the different developments in $\epsilon_N$ and $\epsilon_L$ over time, but it is not clear that they would necessarily generate an anti-correlation. The paragraph has been moved to the LWP sensitivity section and modified to note that "This negative relationship is driven primarily by the different timescales of the LWP and $N_d$ response and could occur even if aerosol produced a strong LWP increase." (P19L2)
* * *
***Page 17, Line 5::*** *"shiptracks are more likely to form in regions with a low cloud top humidity" Can the authors briefly describe a possible mechanism for*

[Figure]

Figure R2: As Fig. 14, but for the instantaneous forcing from each segment.

*this?*

**Reply**: Gryspeerdt et al. (2019) suggested that this could be an enhanced cloud top cooling at low humidity promoting a stronger in-cloud updraught. This makes the cloud less likely to be updraught limited, increasing the sensitivity to aerosol. The sentence has been re-worded to make this clearer. (P18L7)
* * *
***Page 20, Line 5::*** *"this appears to suggest that the increase in sensitivity is almost exactly offset by the decrease in eN" How is this statement derived from? Which figure should the reader refer to?*

**Reply**: This was intended to refer to Fig. 13c, where the $\epsilon_L$ is constant despite a changing sensitivity. The sentence has been modified to reference the "near constant $\epsilon_L$", to make this clearer (P21L3)
* * *
***Figure 14::*** *This is a very useful plot, and I'm also curious how time series of potential radiative forcing (PRF) itself looks like. Can the authors also show them, which should be time derivatives of the integrated forcings shown here?*

**Reply**: The PRF version of Fig. 14 is included below (Fig. R2). We chose the integrated forcing as the instantaneous radiative forcing is very noisy and the integration smooths out some of this noise. In many cases the instantaneous forcing is less than zero, which is unlikely to be a physical response for a large ensemble of tracks (although it may be possible in some isolated cases).
* * *
**Page 22, Line 32::** *"The almost instantaneous LWP adjustment may indicate a retrieval bias" Can the authors briefly discuss how instantaneous negative response of LWP arises from retrieval errors? I don't understand why the DSD-relevant retrieval bias is a potential cause for the negative sensitivity of LWP although discussed in Page 23, Line4.*

**Reply**: Both the LWP and $N_d$ retrievals depend on the cloud optical depth and $r_e$, meaning that they are subject to correlated errors, even if the biases in the optical depth and $r_e$ are random. A random error in the effective radius would lead to a negative correlation between the LWP and $N_d$, generating an artifical negative sensitivity. Although there could be a physical reason for this negative LWP-$N_d$ sensitivity, for such a fast adjustment, the LWP has to change at the speed as the $N_d$ increase, which makes changes in $r_e$ a likely cause. A DSD-related bias in the $r_e$ retrieval is one possible explanation, as changes in the DSD between the clean and polluted cases have been previously observed. This doesn't rule out other options though, particularly the possibility of a physical process causing this near instant LWP adjustment. This section has been re-worded to improve readability. (P24L2)
* * *
**Page 26, Line 27::** *"around 5-10% of clear sky cases in this region are aerosol-limited" Where does this conclusion come from?*

**Reply**: This comes from Fig. 7e and the accompanying discussion (now noted here - P26L9).

**Minor points:**
* * *
**Page 3, Line 5::** *properties (Nd, LWP) shiptracks -¿ properties (Nd, LWP) of shiptracks*
**Reply**: Amended
* * *
**Page 3, Line 26::** *based the method -¿ based on the method*
**Reply**: Amended
* * *
**Page 8, Line 5::** *These -¿ There*
**Reply**: Amended
* * *
**Page 10, Line 13::** *Delete "are"*
**Reply**: Amended
* * *
**Page 12, Line 31::** *microphysical -¿ microphysics*
**Reply**: Amended
* * *
**Page 19, Line 24::** *Delete "a be" prior to "a poor"*
**Reply**: Amended
* * *
**Page 24, Line 4::** *the extent to which -¿ to which extent?*
**Reply**: Modified to "to what extent"
* * *
**Page 25, Line 8:**: *take -¿ taken*
**Reply**: Amended

**Bibliography**

Grosvenor, D. P., Sourdeval, O., Zuidema, P., Ackerman, A., Alexandrov, M. D., Bennartz, R., Boers, R., Cairns, B., Chiu, J. C., Christensen, M., Deneke, H., Diamond, M., Feingold, G., Fridlind, A., Hünerbein, A., Knist, C., Kollias, P., Marshak, A., McCoy, D., Merk, D., Painemal, D., Rausch, J., Rosenfeld, D., Russchenberg, H., Seifert, P., Sinclair, K., Stier, P., van Diedenhoven, B., Wendisch, M., Werner, F., Wood, R., Zhang, Z., and Quaas, J.: Remote Sensing of Droplet Number Concentration in Warm Clouds: A Review of the Current State of Knowledge and Perspectives, Rev. Geophys., https://doi.org/10.1029/2017RG000593, 2018.

Gryspeerdt, E., Goren, T., Sourdeval, O., Quaas, J., Mülmenstädt, J., Dipu, S., Unglaub, C., Gettelman, A., and Christensen, M.: Constraining the aerosol influence on cloud liquid water path, Atmos. Chem. Phys., 19, 5331–5347, https://doi.org/10.5194/acp-19-5331-2019, 2019.

Yuan, T., Wang, C., Song, H., Platnick, S., Meyer, K., and Oreopoulos, L.: Automatically Finding Ship Tracks to Enable Large-Scale Analysis of Aerosol-Cloud Interactions, Geophys. Res. Lett., https://doi.org/10.1029/2019GL083441, 2019.